

# Dynamics of carbon loss from an arenosol by a forest/vineyard land use change on a centennial scale

Solène Quéro[1*], Christine Hatté[2], Sophie Cornu[1], Adrien Duvivier[1], Nithavong Cam[1], Floriane Jamoteau[1,3], Daniel Borschneck[1], Isabelle Basile-Doelsch[1]

[1]Aix Marseille Univ, CNRS, IRD, INRAE, CEREGE, Aix-en-Provence, France
[2]Laboratoire des Sciences du Climat et de l'Environnement, UMR 8212 CEACNRSUVSQ, Université Paris-Saclay, Gif-sur-Yvette, France.
[3]CIRAD, Internal Research Unit (UPR) Recycling and Risk, Station de La Bretagne, Réunion, France

*Correspondence to*: Solène Quéro (quero@cerege.fr)

**Abstract.** Few studies have focused on arenosols with regard to soil carbon dynamics despite the fact that they represent 8% of the world's soils and are present in key areas where food security is a major issue (e.g. in Sahelian regions). As for other soil types, land use changes (from forest or grassland to cropland) lead to a loss of substantial soil organic carbon (SOC) stocks and have a lasting impact on the SOC turnover. Here we quantified long-term variations in carbon stocks and their dynamics in a 80 cm deep Mediterranean Arenosol that had undergone a land use change from forest to vineyard over more than 100 years ago. Paired-sites of adjacent plots combined with carbon and nitrogen quantification and natural radiocarbon ($^{14}$C) abundance analyses revealed a stock of 50 GtC ha$^{-1}$ in the 0-30 cm forest soil horizon, which was reduced to 3 GtC ha$^{-1}$ after long-term grape cultivation. TOC in vineyard was dramatically low, with around 1 gC kg$^{-1}$ and no vertical gradient as a function of depth. $^{14}$C showed that deep ploughing (50 cm) in vineyard plot redistributed the remaining carbon both vertically and horizontally. This remaining carbon was old carbon (compared to that of the forest), which had a C:N ratio characteristic of microbial OM and was probably stabilized within organomineral associations. Despite the drastic degradation of the OM pool in this Arenosol, this soil would have a high carbon storage potential if agricultural practices, such as grassing or organic amendment applications, were to be implemented within the framework of the 4 per 1000 Initiative.

## 1 Introduction

Arenosols account for 8% of the world's soils and are found mostly under desert, tropical and Mediterranean climatic conditions (FAO, 2014). They are silty-sandy or sandy soils, with less than 35% by volume of coarse elements and occur in layers about 100 cm deep (FAO, 2014). Surface carbon concentrations of arenosols range from 100 g kg$^{-1}$ for the richest (Andreetta et al., 2013) to 1 g kg$^{-1}$ for the poorest (Fourie et al., 2005; López-Piñeiro, 2013), with stocks in the 0-30 cm layer ranging from 15 tC ha$^{-1}$ (Muñoz-Rojas et al., 2012) to more than 80 tC ha$^{-1}$ (Marschner and Waldemar Wilczynski, 1991), with 80 tC ha$^{-1}$ being the average for global soils (Mousset, 2014). As with other soil types, changes in arenosol land use patterns



(from forest or grassland to cropland) can lead to a loss of carbon (Lal, 2004). Arenosols can also be considered as soils with storage potential suitable for meeting targets of the 4 per 1000 Initiative (Lal, 2004; Minasny et al., 2017). As these soils are present in key areas for future food production (FAO, 2018), understanding carbon dynamics in arenosols is therefore also a crucial societal challenge. Unfortunately, however, few studies to date have focused on this type of soil (Kögel-Knabner and
Amelung, 2021).

Whatever the soil type, land use changes (forest or grassland to cropland) may lead to high and rapid loss of carbon by erosion, runoff and/or mineralisation ($CO_2$ release), of about 50% in 10 years (Guillaume et al., 2021; Ramesh et al., 2019). Most of this loss occurs in the surface horizons (0-30 cm), which are very sensitive to disturbance (Lal, 2004), which explains why most studies generally focus on topsoils (Baker et al., 2007). Land use changes also affect deep stocks representing a significant
amount of carbon, although quantitatively less than at the surface (Angers and Eriksen-Hamel, 2008; Basile-Doelsch et al., 2009; Poeplau and Don, 2013; Wang et al., 1999). Therefore, it would be essential to gain insight into the dynamics at the profile scale, not only at the agronomic horizon scale.

Institutional experimental sites with temporal monitoring have plots that are highly suitable for studying the impact of cultivation. However, experiments lasting more than 50 years are uncommon and generally there are no equivalent plots in
old-growth forests (i.e. where the soil has not been disturbed by cultivation for at least 100 years). Apart from these experimental sites, the paired-site study strategy is highly relevant (Eldon and Gershenson, 2015). A site pair is defined as two plots with different uses on the same soil, under the same climatic conditions, on the same bedrock, and on a flat topography. However, as these conditions are difficult to meet, few studies have been carried out on pairs of soils in strict compliance with the above criteria, let alone over a long period of time to assess significant differences in carbon content between cultivated
and forest soils. In the metanalysis of Eldon & Gershenson (2015), for example, the study times did not exceed 50 years. Balesdent et al. (2018) studied paired sites with a change in vegetation from $C_3$ to $C_4$, or vice versa, to assess the age of deep carbon stocks. This is an efficient method but only applicable to specific conditions (difference in [13]C isotopic signature between two successive vegetation types). Otherwise [14]C may be applicable to any system to assess the impact of cultivation on carbon dynamics at the decadal (or longer) scale as it is a function of carbon age (Trumbore, 2009). As early as 1993, a few
researchers used [14]C in a land use change context (Harrison et al., 1993; Trumbore, 1993), but since then few studies have followed suit. For example, only 3 of the 789 soil profiles documented in papers in the International Soil Radiocarbon Database (Anon, 2020; Lawrence et al., 2020) mention land use change (Dümig et al., 2008; James et al., 2019; Monreal et al., 1997). Cultivation mainly affects young (short turnover) carbon pools at the surface by promoting their mineralisation, but more stable (long turnover) carbon pools may also be impacted via their transfer to carbon pools with faster turnover (Poeplau
and Don, 2013), thus leading to overall ageing of soil organic matter (OM), at least at the surface (Wang et al., 1999).

Land use change affects carbon stocks and dynamics, but agricultural practices also have an effect. Surface (0-30 cm) or deep (> 50 cm) ploughing, in particular, as carried out in vineyards, disturbs the vertical and horizontal distribution of carbon (Dimassi et al., 2014; Mary et al., 2020). [14]C is often used in studies to assess carbon dynamics in soils. However, few authors refer to the heterogeneity of the [14]C content within the same layer  (van der Voort et al., 2016). Only two paper (Chiti et al.,

2016; van der Voort et al., 2016) in the International Soil Radiocarbon Database (Anon, 2020; Lawrence et al., 2020) dealt with single-layer heterogeneity and neither of them considered cultivated soils. The main obstacle to the use of $^{14}$C is the high cost of analysis, which often warrants a single measurement. Samples can be pooled into a single composite sample to overcome this heterogeneity problem (Jiang et al., 2020). This is why—despite being essential in a tillage context—few studies to date have focused on intra-horizon variability.

Finally, although arenosols lose a lot of carbon during cultivation (Fourie et al., 2005; López-Piñeiro, 2013), they have a high storage potential. Experiments implementing storage practices on arenosols (Kazlauskaite-Jadzevice et al., 2019) documented an increase of 40.2 to 45.6 t ha$^{-1}$ (+ 13.4 %) and 39.4 to 49.0 t ha$^{-1}$ (+ 24.4 %) in soil organic carbon stock (SOC stock) in the 0-30 cm soil layer in 20 years following cropland abandonment and a grassland management change, respectively. Arenosols, which have very low carbon stocks in cultivated systems, thus seem to be good candidates for the 4 per 1000 (4p1000)

Initiative.

   This study was therefore carried out to highlight the impact of the establishment and management of a vineyard on an arenosol, as well as its consequences on the carbon dynamics at soil layer and entire soil profile scales. It also aimed to highlight links between the degree of biotransformation of OM by the microbial compartment and its age, and finally, to discuss the storage potential of arenosols with respect to the 4p1000 criteria. We thus based our study on paired soils, measuring carbon contents

and stocks, vertical and intra-horizon heterogeneity of carbon as measured by $^{14}$C, and correlating the C:N ratio and radiocarbon (F$^{14}$C). Finally, we applied a rate of carbon incorporation in our cultivated arenosol according to the proportions and rate put forward in the remediation study of Kazlauskaite-Jadzevice et al. (2019).

## 2 Materials and methods

### 2.1 Study area

The study site was located at Plan de la Tour, in the Maures massif (France), under a Mediterranean climate:   -3°C < T$_{Winter}$ < 18°C and T$_{Summer}$ > 22°C, P$_{driest month}$ < 40 mm (Rubel and Kottek, 2010). The soil was a poorly differentiated arenosol on granite. The site consisted of two plots on a flat landscape: one in a forest and the other in a vineyard. An analysis of aerial photographs and cadastral maps (from 1813 to present day) showed that these two plots had a history of continuous soil use for at least ~100 years in the case of the forest (with an age of 91 years, as measured by dendrochronology on a cork oak) and

more than 150 years in the case of the vineyard (Fig.B1). Additional field work ruled out the effects of terracing at the selected sampling spots. The vineyard plot had undergone vine uprooting and deep ploughing (~50 cm) every 70 years on average. The last ploughing was carried out between 1998 and 2003. The soil was bare between rows (Fig.1 and Fig.B1).





## 2.2 Sampling

Two pits were dug down to the underlying granite: the forest pit (43°19'37.35 "N, 6°32'12.89 "E) was 70 cm deep and the vineyard pit (43°19'37.74 "N, 6°32'11.90 "E) was 80 cm deep (Fig.1). The pits were 15 m apart. The grain size and mineralogy were similar at both sites (Fig.B2 and Fig.B3). Three faces were sampled per pit (A, B and C). Soil cylinders were taken from the three faces in the vineyard pit (100 ml) and from two faces in the forest pit (1230 ml, only below 20 cm) to determine the bulk density. Above 20 cm, in the forest soil, the water measurement technique was preferred to the cylinder technique due to

the high abundance of tree roots. Bulk density samples were oven dried at 105°C for 3 days before weighing. The profile samples were air-dried (25°C) for 1 week, sieved (2 mm) and ground in a planetary mill (50 g for 5 min, including 1 min reverse, at 400 rpm) down to <200 µm and quartered. For the $^{14}$C analysis, a 3 g composite sample (i.e. a mix of 1 g of A, B and C) was prepared for each depth range. To test the intra-horizon variability, 5-10 cm and 40-50 cm samples in the forest and vineyard, and also 50-60 cm samples in the vineyard (below the ploughing sole) were selected for further analyses (Fig.1,

Table C1 and C2). This variability was used to extrapolate the variability at all depths for in the vineyard and forest.

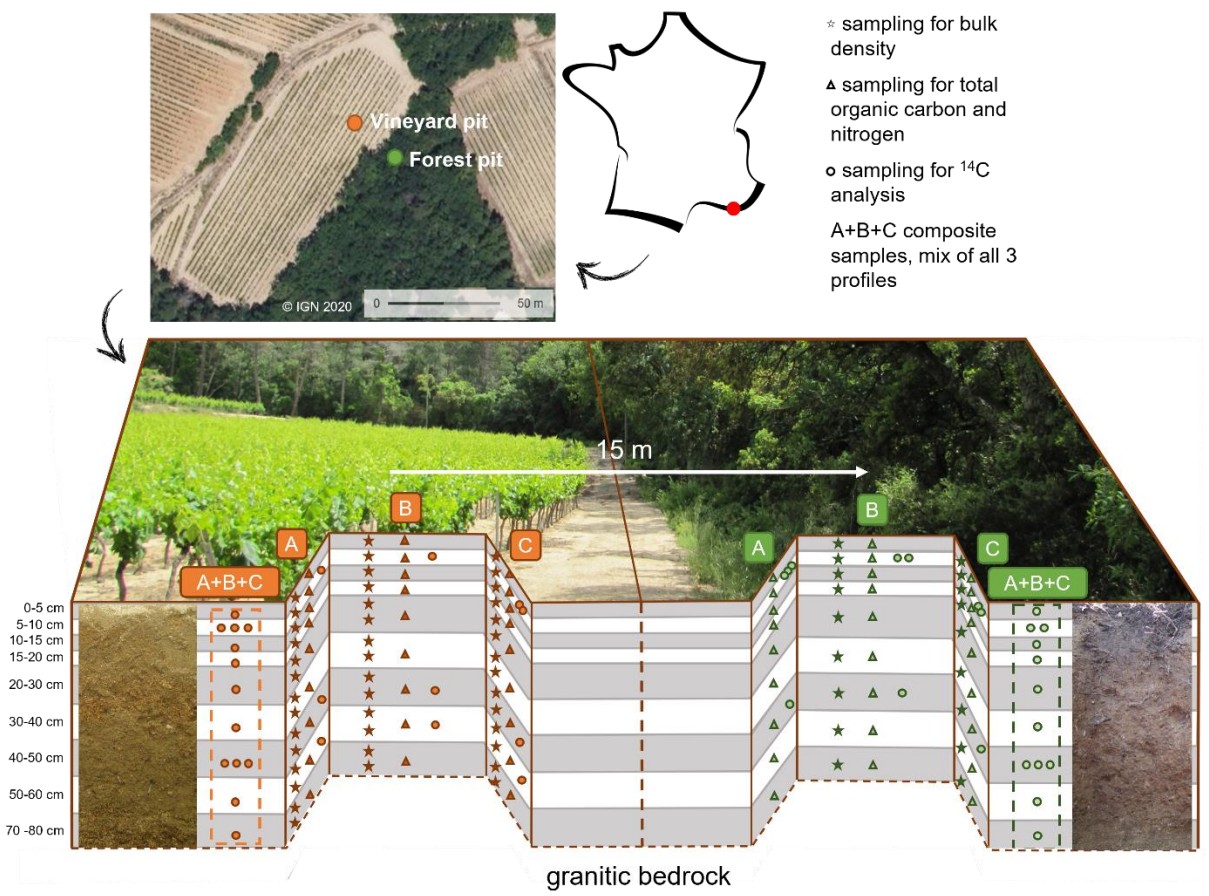





**Figure 1: Scheme of the pits and sampling in the Brugassières arenosol under a Mediterranean climate: left, vineyard site (orange symbols); right, forest site (green symbols). A, B and C represent the three different sampled sides of each pit. Symbols indicate the**
**sampling and analysis for each sampled layer: stars, sampling in cylinders for bulk density; triangles, sampling for total organic carbon and total nitrogen (TOC, TN), granulometry and mineralogy; circles, sampling for analysis in 14C; (A+B+C) represent composite samples resulting from the mixture of samples from the three faces at equal proportions. The aerial photo was from Géoportail website ([https://www.geoportail.gouv.fr](https://www.geoportail.gouv.fr)), the other photos were from personal sources.**

## 2.3 Methods

### 2.3.1 Carbon and nitrogen measurement, and stocks calculation

Carbon and nitrogen contents were measured by dry combustion with an elemental analyser (NF ISO 10694 and 13878, respectively). Soil organic carbon stocks (SOC stock) were calculated according to Eq. (1):

$$SOC\ stock = BD * TOC * e * 10 \tag{1}$$

Carbon stocks are expressed in t·ha$^{-1}$, carbon content (TOC) in g·kg$^{-1}$, bulk density (BD) in g· cm$^{-3}$ and layer thickness (e) in
cm. A correction (equation 2) was then applied in order to compare carbon stocks at equivalent mass and thus eliminate differences in bulk density between the two sites for the same depth (Ellert and Bettany, 1995; Poeplau et al., 2017; Poeplau and Don, 2013) Eq. (2) :

$$SOC\ stock_{cor} = SOC\ stock - \left(\frac{M_{sample} - M_{eq}}{V} * TOC * e * 10\right) \tag{2}$$

Here V is a predefined volume of 1 cm$^3$, $M_{sample}$ is the sample mass in g, with $M_{sample} = BD*V$ and $M_{eq}$ being the equivalent
mass in g, which corresponds to the mass of the least dense sample (i.e forest sample) between the two compared sites, for the same depth.

### 2.3.2 Carbon and nitrogen measurement, and stocks calculation

The sample $^{14}$C contents were evaluated on raw samples (no chemistry applied). The most carbon-rich samples (forest <40 cm) were measured on the solid source of the ECHoMICADAS (Synal et al. 2007, Tisnérat-Laborde et al., 2015) or via the gas
source. Organic carbon of the carbon-rich samples was first transformed into graphite on AGE3 (Wacker et al., 2010). The analyzed carbon quantity ranged from 24 to 86 µg of carbon in the vine profiles samples, 87 to 89 and 983 to 1000 µg of carbon in the forest profiles samples measured through the gas source and the solid source, respectively. The range of variation of the analytical error, expressed as F$^{14}$C, was between 0.002 and 0.014 and decreased with increasing carbon mass (Fig.E1). The radiocarbon data are expressed in modern F$^{14}$C fraction, as recommended by Reimer et al. (2004). The difference between
the highest and lowest F$^{14}$C values for the same depth is expressed by ΔF$^{14}$C. However, since many authors have used the Δ$^{14}$C or conventional radiocarbon age to express $^{14}$C (Reimer et al., 2004; Stuiver and Polach, 1977), the data expressed in Δ$^{14}$C and conventional age are shown in Fig.E2 to facilitate comparison. All equations for the different units can be found in appendix A.





## 3 Results

### 3.1 Carbon content, C:N ratio and stocks

The results of the carbon content profiles are presented in Fig.2a. Under the forest, carbon content and variability were high at the surface, with 33-56 g kg$^{-1}$ in the 0-5 cm layer ($\pm$ 9), but it decreased with depth down to 1.89-2.50 g kg$^{-1}$ in the 0-60 cm layer ($\pm$ 0.06). Under vines, the carbon content was comparatively very depleted and equivalent in the 3 profiles throughout the depth. At the surface (0-5 cm), the TOC ranged from 0.9 to 2.4 g kg$^{-1}$ ($\pm$ 0.8), and at depth (70-80 cm) it ranged from 0.34 to 0.82 g kg$^{-1}$ ($\pm$ 0.29). The TOC values under vines were extremely low compared to those under the forest (23-fold lower than under the forest in the 0-5 cm layer) and this depletion was even observed at depth. The C:N ratios are presented in Fig.2b. Under the forest, the average C:N ratio was high, i.e. around 16 in the 0-5 cm layer, and decreased with depth to 10 in the 60-70 cm layer. The C:N ratio under vines was twice as low as noted in the forest surface horizon (8 $\pm$ 3 versus 16 in the 0-5 cm layer), and there were variations in this ratio up to 50 cm depth. Beyond this depth, the vineyard profile became similar to that under the forest. Finally, the cumulative stocks at 30 cm depth are represented in Fig.2c, and the forest soil contained 50 tC ha$^{-1}$ while the vineyard soil contained only 3 tC ha$^{-1}$.

### 3.2 Radiocarbon

The radiocarbon profile results are presented in Fig.2d. Young carbon was detected in the forest profile, i.e. younger than the 1960 bom peak ($F^{14}C > 1$), at the surface. The carbon age then increased with depth ($F^{14}C < 1$ around 40 cm). This was a conventional undisturbed soil profile (Jreich, 2018; Mathieu et al., 2015; van der Voort et al., 2016), which shows a 'belly' shape curve between 5 and 20 cm depth. This belly shape curve corresponded to the penetration of the $^{14}C$ signal of the bomb peak in the profile. Concerning the variability in a single soil layer, $F^{14}C$ ranged from 1.095 to 1.124 ($\Delta F^{14}C = 0.029$) at the surface (5-10 cm). Meanwhile, at depth (40-50 cm), $F^{14}C$ ranged from 0.974 to 1.005 ($\Delta F^{14}C = 0.031$; Fig.2d).

Conversely, the vineyard profile revealed the presence of old carbon from the surface to the bottom of the pit ($F^{14}C = 0.893$ at the surface and 0.990 at depth), despite the heterogeneity within the horizons (one point with an $F^{14}C > 1$, at 40-50 cm). The variation pattern in the profile was not progressive from the soil surface to the depth, contrary to the pattern noted in the forest profile. Under vines, the intra-horizon variability was much more marked than under the forest. In the 0-10 cm layer, $F^{14}C$ ranged from 0.880 to 0.969 ($\Delta F^{14}C = 0.089$), and from 0.909 to 1.081 ($\Delta F^{14}C = 0.172$) at 40-50 cm depth.





165

**Figure 2: TOC variations (a), C:N ratio (b), cumulative stocks corrected for equivalent mass (c) and F¹⁴C profiles (d) as a function of depth, under vines (orange) and under forest (green), for an arenosol under a Mediterranean climate. The F¹⁴C measurement variability (d) is represented by green (forest) and orange (vine) bands.**





## 4 Discussion

### 4.1 Comparison with Mediterranean arenosols

Under the forest, the TOC profiles ($22 \pm 5$ g kg$^{-1}$ in the 0-20 cm layer and $3.94 \pm 0.25$ g kg$^{-1}$ in the 30-50 cm layer) obtained for topsoil and subsoil were comparable to those obtained for other arenosols under Mediterranean climatic conditions (Figure 3a) (Andreetta et al., 2013; Caravaca et al., 2002; Fierro et al., 2007; Pinzari et al., 1999; Vittori Antisari et al., 2016). However, to our knowledge, very few data are available beyond 30 cm soil depth. For comparison, we only identified 5 references of studies concerning arenosols under grapevines in Mediterranean climatic conditions. The Brugassières arenosol was found to be among the soils with the lowest of organic carbon content values (Figure 3b). This trend was visible in surface soils as well as at depth. Some arenosols under vines reportedly had low carbon contents comparable to those in the soil studied here, both at the surface and at depth (Fourie et al., 2005; López-Piñeiro, 2013).The arenosol in this study, although very depleted in C, does not seem to represent a unique case of OM depletion after arenosol vineyard cultivation.

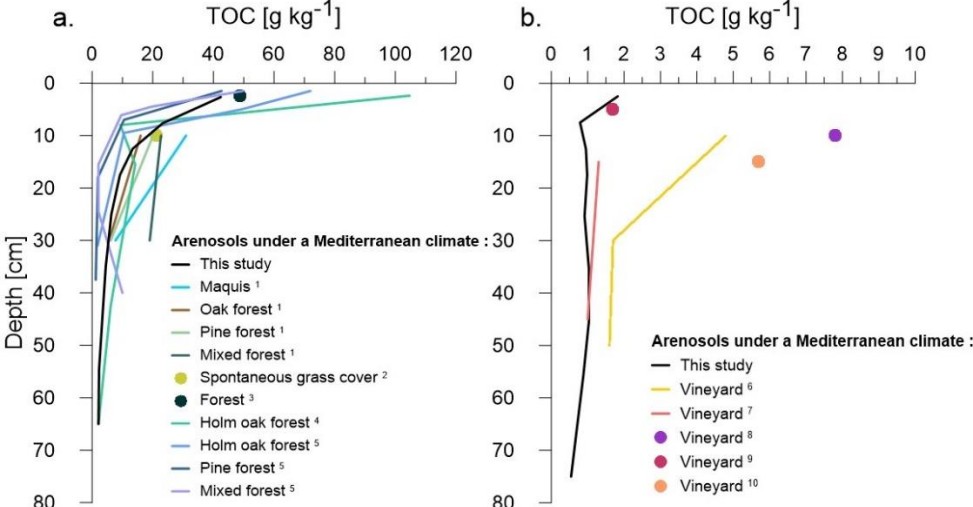

**Figure 3: TOC comparison between the Brugassières arenosol and other forest arenosols (a) and vineyard arenosols (b) under a Mediterranean climate (1) Pinzari et al., (1999); (2) Caravaca et al., (2002) ; (3) Fierro et al., (2007) ; (4) Andreetta et al., (2013) ; (5) Vittori Antisari et al., (2016) ; (6) Conradie, (2001) ; (7) Fourie et al., (2005) ; (8) Okur et al., (2009) ; (9) López-Piñeiro, (2013) ; (10) Nogales et al., (2019). Data available in Table D1.**

### 4.2 Drastic carbon stock loss: a combination of land use change / agricultural practices / unfavourable soil texture

These very low carbon contents in the vineyard resulted in a 12-fold lower carbon stock in the vineyard than in the forest throughout the profile (e.g. in the 0-30 cm layer, the SOC stock was 3.2 t ha$^{-1}$ in the vineyard compared to 50.8 t ha$^{-1}$ in the forest) (Fig.2c). Arenosol carbon stocks under the forest, in the 0-30 cm layer, were lower than stocks under the forest irrespective of the soil type (80 t ha$^{-1}$, Mousset, (2014)). The difference between the national forest average and that of the studied forest was: $80 - 50.8 = 29.2$ t ha$^{-1}$. This suggests that arenosol carbon stocks under vines were lower than the average in French vineyards (30 t ha$^{-1}$, Mousset, 2014). The difference between the national mean and that of the studied vineyard was:





30–3,2 = 26.8 t ha$^{-1}$. Consequently, arenosols had about 30 tC ha$^{-1}$ less than the French average regardless of the soil type under both forests and vines.

Cultivation in the vineyard plot resulted in a very high carbon stock loss throughout the entire depth: 94% in the 0-30 cm layer and 76% in the 30-60 cm layer. Although this carbon stock loss phenomenon has already been widely reported (Guillaume et al., 2021; Ramesh et al., 2019), it has generally been found to be around 50% at the surface during a forest (or grassland) to vineyard transition under all climatic conditions (Carlisle et al., 2006; Eldon and Gershenson, 2015). Moreover, contrary to our findings here, the loss is usually much greater in topsoil than in subsoil layers, ranging from 30 to 63% on average in the 30-100 cm horizon (Batjes, 2014; Poeplau and Don, 2013). However, if we focus the comparison on arenosols under a Mediterranean climate, losses (in TOC) during a natural vegetation to vine transition can reach 85% in the 0-20 cm layer over a 1 year period (Caravaca et al., 2002). The soil carbon loss noted in this study thus resulted in an extremely high carbon loss after more than 150 years of grapevine cultivation, which does not seem to be out of line with observations described in the literature.

This extreme carbon loss throughout the cultivated soil profile could be explained by a combination of four aggravating factors at the Brugassières site: (1) The initial disturbance of the arenosol, due to the forest to vineyard land use change in the 19th century (Caravaca et al., 2002; Tsozué et al., 2020); (2) The absence of vegetation cover (apart from vines) for more than 150 years was probably also an important factor. Carbon inputs were almost nil at the surface (soil kept bare, Fig.1). Deep inputs were limited to the depth of the grapevine root system, while the vine plants were uprooted every 70 years. However, the age of the carbon distribution as a function of depth proposed by Balesdent et al. (2018) shows that almost half of the carbon in a soil is on average younger than 150 years at the soil profile scale. Although this distribution concerns soils under tropical climates, the drastic long-term reduction of carbon inputs to the soil could likely largely explain the carbon stocks observed in the vineyard throughout the soil profile; (3) Deep ploughing (50 cm), carried out every 70 years at the same time as the grapevine plant uprooting, was probably a third factor favouring carbon loss via accelerated SOC mineralisation; and finally (4) The arenosol texture, characterised by a low proportion of fine particles (< 20 µm fraction, Fig.B2) is also an unfavorable factor for C storage within the mineral-associated OMs.

### 4.3 Intra-layer radiocarbon variability

Carbon spatial heterogeneity is generally not taken into account in soil studies on carbon dynamics using the $^{14}$C proxy (van der Voort et al., 2016). We only found two papers listed in the Shi et al. (2020) database on forest and cultivated soils that addressed intra-layer variability, i.e. Chiti et al. (2016) and van der Voort et al. (2016). These authors showed that the intra-layer radiocarbon signature under forests is relatively homogeneous at the soil surface and at depth. This finding is in line with our forest soil results (Fig.2d, Fig.4), where the low intra-layer $F^{14}C$ variability (represented by the standard deviation, SD) in the forest soil was noted both in the 5-10 cm layer with a high carbon concentration ($SD_{F14C} = 0.008$; $TOC_{average}$=42.4 gC kg$^{-1}$ and $SD_{TOC}$= 9.1 gC kg$^{-1}$) and in the 40-50 cm layer with a low carbon concentration ($SD_{F14C} = 0.011$; $TOC_{average}$= 3.4 gC kg$^{-1}$ and $SD_{TOC}$= 0.1 gC kg$^{-1}$). Low intra-layer variability was also observed in the vineyard soil (Fig.2d, Fig.4), below the ploughing



depth (50-60 cm layer, $SD_{F14C}$ = 0.012; $TOC_{average}$ = 0.9 gC kg$^{-1}$ and $SD_{TOC}$ = 0.1 gC kg$^{-1}$). Thus, in a horizon undisturbed by agricultural cultivation, $^{14}$C showed little intra-layer variability on a metric scale, even when measurements were carried out on samples with a very low TOC (Fig.E3).

In cultivated systems, to our knowledge, no studies have reported measurement of intra-layer $^{14}$C variability. Our findings therefore cannot be compared with those of previous studies. In comparison to undisturbed horizons, much higher intra-layer
variability was observed at 5-10 cm depth ($SD_{F14C}$ = 0.029) and within the 40-50 cm ploughing depth ($SD_{F14C}$ = 0.058), with both layers being characterized by low total carbon (over 5-10 cm, TOC= 0.79 gC kg$^{-1}$ with SD=0.26 gC kg$^{-1}$; over 40-50 cm, TOC= 1.03 gC kg$^{-1}$ with SD= 0.42 gC kg$^{-1}$). Furthermore, the F$^{14}$C measurements at 40-50 cm depth in the B profile and in vineyard pit composite soils had a post-bomb value (F$^{14}$C$_{mean}$ = 1.001), which was higher than that obtained in the forest soil (F$^{14}$C$_{mean}$ = 0.990). At 40-50 cm depth, and only for this horizon, OM in the vineyard was therefore younger than that in the
forest soil. This highly suggests that the variability in F$^{14}$C measured between the samples on sides A, B and C was a consequence of multiple ploughing whereby the soil is mixed vertically but also horizontally on a metric scale.



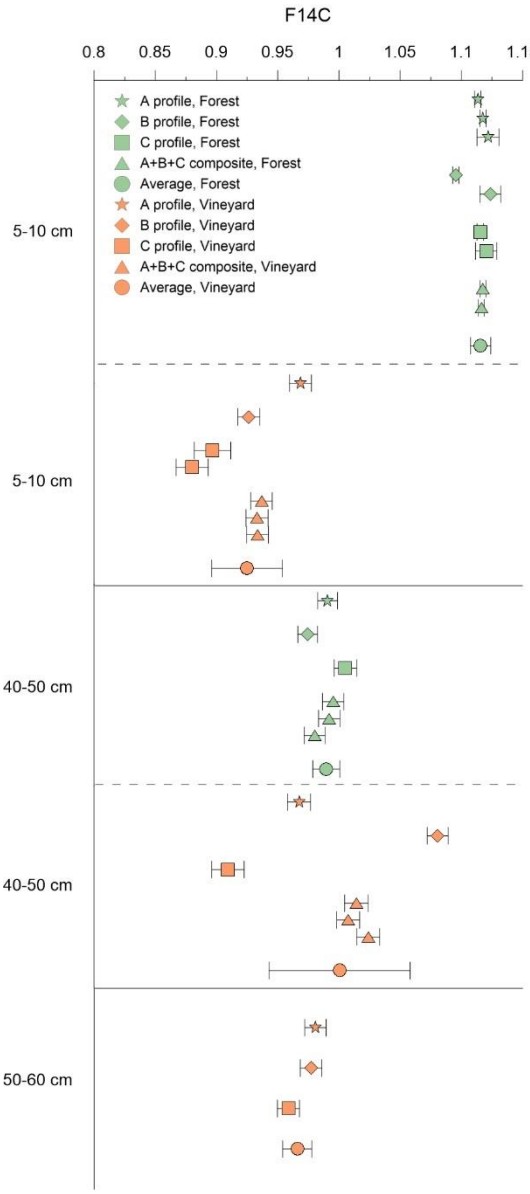

**Figure 4: Comparison of intra-layer F¹⁴C heterogeneity at three depths (5-10, 40-50 and 50-60 cm) in forest and vineyard soils. F¹⁴C**
**data were obtained for profiles A (star), B (diamond), C (square), composites A+B+C (triangle) and the average of these data (round),**
**in forest (green) and vineyard (orange) soils. Error bars represent the analytical error for the profiles A, B and C and the standard**
**deviation for the mean.**

**4.4 Intra-layer radiocarbon variability**

Based on the meta-analysis of (Shi et al., 2020) and version 1.7.8.2021-01-04 of the ISRaD database (ISRaD, 2020; Lawrence

et al., 2020), Figure 5 compares the $\Delta^{14}C$ of soil profiles reported in 185 papers, under forest and cultivation, at the soil surface

(0-30 cm) and at depth (30-100 cm). The arenosol studied here had a higher $\Delta^{14}C$ than the median $\Delta^{14}C$, all soil types combined,



in topsoil and even more marked in subsoil layers, e.g. in topsoil, the $\Delta^{14}C_{forest}$= 91.5 ±166.8, relative to a median of -9 (-79; 46) from the literature; and $\Delta^{14}C_{crop}$= -32.7 ±183, relative to a median of -58 (-171; -7). This was probably due to the lower fine particle content ($< 2\,\mu m$) than the overall average in the meta-analysis. Indeed, arenosols have few reactive mineral phases

that stabilise OM in the long term, which is in line with the above discussion on stocks. The fact that the OMs were systematically younger than those generally described in the literature could thus be explained by the soil type (i.e. the fine fraction was minimal in the arenosol) and by the long cropland history (> 150 years).

### 4.5 Land use impact on OM borne $^{14}C$

In the ploughed horizon, with the exception of the 40-50 cm layer, $\Delta^{14}C$ was always more negative in cultivated soils than in
forest soils. Cultivation therefore led to carbon aging (by loss of the most recent carbon pool) to 40 cm depth. This impact of cultivation had already been highlighted in a ploughed horizon by Wang et al. (1999), where the carbon of a cultivated soil in the 0-30 cm layer was older than its equivalent in forest soils. This trend was also revealed in a meta-analysis (Figure 5, cropland-soil n=34, forest-soil, n=151 papers). The median values confirmed that the carbon age of SOM was older in cultivated soils in both the surface and deep horizons. Cultivation affects the mean carbon turnover by mainly removing carbon
from fast-turnover pools and retaining mostly slow-turnover carbon pools (Poeplau and Don, 2013). It is likely that these OM pools are organic compounds associated with the mineral-associated OM (MAOMs) pool (Cotrufo et al., 2019) . Furthermore, the findings in the 40-50 cm horizon, with a younger post-bomb OM than all other horizons in the vineyard profile, showed that the full inversion tillage practiced effectively dragged surface OM down to 50 cm. Cultivating the deep ploughed arenosol under vines therefore led to (1) loss of the young and poorly stabilised OM pools, and (2) redistribution of the remaining
MAOMs throughout the ploughed horizon and, as shown in section 4.3, in a horizontally heterogeneous way.

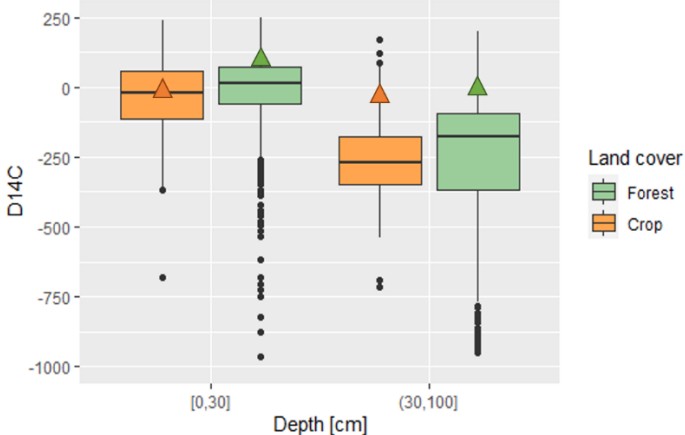

**Figure 5: Comparison of the $\Delta^{14}C$ average at 0-30 cm and 30-100 cm depth between this study forest (green triangle), this study crop (orange triangle) and the $\Delta14C$ average from the global ISRaD database (Lawrence et al., 2020; Shi et al., 2020). Black dots represent outliers. Crop-soil n=34, forest-soil n=151 papers.**






#### 4.6 Microbial origin of OM in the vineyard

The C:N ratio of the soil under forest (e.g. 13 < C:N <16 on the 0-20 cm) was consistent with the average for world soils, (between 9.9<C:N<25.8 (Batjes, 2014)) and with the average for deciduous forest soils (C:N = 13.8 ± 4.0 (Cotrufo et al., 2019)). These values mainly corresponded to OM of plant origin. The soil C:N ratio under vines (7<C:N<12) corresponded to

the OM C:N ratio mainly of microbial origin associated with minerals (C:N$_{MAOM}$= 12.6 ±4.7, Cotrufo et al., 2019). The positive correlation obtained between the C:N ratio and F$^{14}$C (Fig.6) confirmed that the ancient carbon present in the soil was mainly borne by molecules originating from N-rich microbial metabolism and presumably stabilised within MAOMs (Cotrufo et al., 2019; Kleber et al., 2015). The change in vineyard use associated with conventional practices (absence of inter-row cover crops and deep ploughing) thus seems to only allow the maintenance of this small pool of MAOMs, to the detriment of other

less stable OM pools, lost through erosion, leaching or mineralisation.

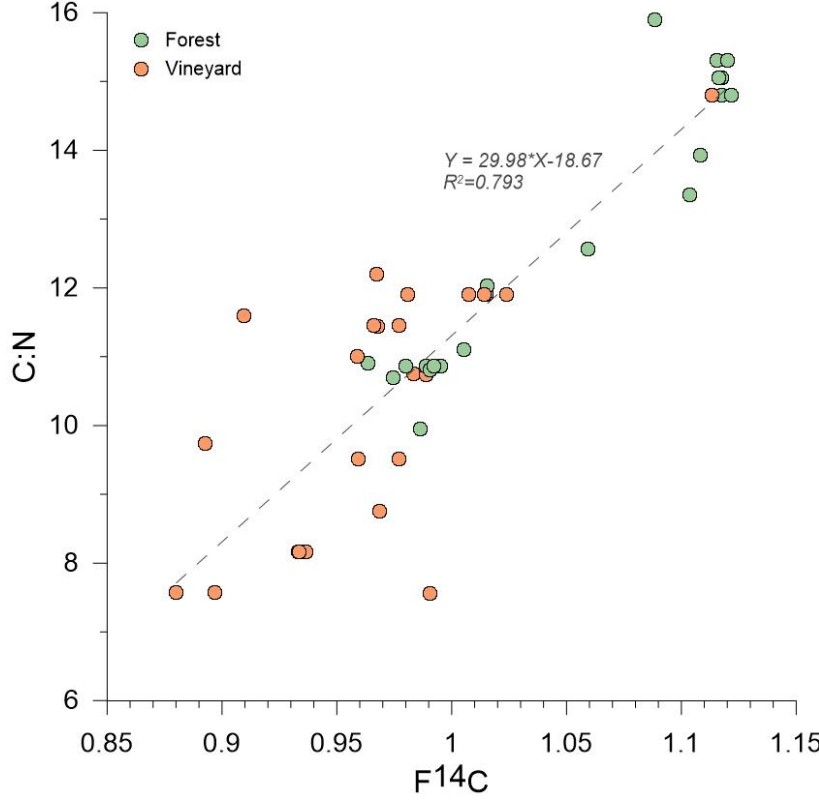

**Figure 6: Correlation between the F$^{14}$C and C:N ratio. The correlation was calculated on composite samples (F$^{14}$C) and the average for the 3 profiles A, B and C (C:N), as well as on samples of profiles A, B and C only, from the forest (green) and vineyard (orange).**





### 285 4.7 Are arénosol a good target for the 4:1000 Initiative?

To restore OM stocks in soils and meet the 4p1000 objectives, the land use pattern may be changed (cropland returned to grassland or forest), or cropland may be maintained by adopting practices that foster C storage, e.g. establishment of permanent grasslands, application of organic amendments, grassing of vineyards, etc. (Pellerin et al, 2019). Arenosols, whose carbon stocks are very low in cultivated systems (Fig.2c), thus seem to be good candidates for the 4p1000 Initiative because they have

a high C storage potential. Storage experiments conducted on arenosols measured an increase of 40.2 to 45.6 t ha$^{-1}$ and 39.4 to 49.0 t ha$^{-1}$ of carbon stocks in the 0-30 cm layer in 20 years following, respectively, cropland abandonment (+ 0.27 t ha$^{-1}$ yr$^{-1}$) and a change of grassland management (+ 0.48 t ha$^{-1}$ yr$^{-1}$) (Kazlauskaite-Jadzevice et al., 2019). In these experiments the annual increase in carbon stock was +5.9 % and +9.8% of the final stock, respectively, i.e. more than 10-fold higher than the 0.4 % annual increase targeted by the 4 per 1000 Initiative.

In the case of the studied arenosols, the potentially achievable reference stock could be considered equal to the forest soil stock. In the 0-30 cm range, the C storage potential was therefore 48 t ha$^{-1}$ (Fig.2c). If we consider an annual C stock increase rate equivalet to that obtained by Kazlauskaite-Jadzevice et al. (2019), we could calculate that an arenosol could recover this stock in 14 years under appropriate practices (i.e. 3.4 t ha$^{-1}$ yr$^{-1}$). If we calculate differently, considering not the same storage proportion as Kazlauskaite-Jadzevice et al. (2019) but the same storage rate (a mean value of + 0.37 t ha$^{-1}$ yr$^{-1}$), an arenosol

could recover its C stock in 128 years of storage practice. The reality would probably fall between these two values and the system would probably not respond linearly. The additional C storage potential in cultivated arenosols is thus high, but these calculations, although very simplistic, show that there is still considerable uncertainty about the time needed to reach maximum storage. In any case, although these time scales are still poorly understood, ambitious annual soil carbon storage objectives could clearly be met in arenosols upon the adoption of C storage practices.

### 305 5 Conclusion

Land use change from a Mediterranean forest to a vineyard on an arenosol resulted in loss of almost all of the soil's carbon throughout the entire depth of the soil profile: 93.7% less SOC at the surface and 76.2 % at depth. The few research papers (2) that we found with comparable levels did not report enough detail (e.g. history of the plots) to be able to understand why their reported values corresponded to those found at our Les Brugassières site. The radiocarbon study highlighted the very high

vertical homogeneity (as a function of depth) and horizontal heterogeneity (intra-layer) of the carbon distribution, induced by the deep ploughing practice. The carbon remaining in the >50 cm soil layer was old, stabilized microbial carbon that was mixed with younger carbon at depth. The study of [14]C data and the C:N ratio revealed a link between the degree of OM biotransformation by the microbial compartment and its age, i.e. F[14]C (old and stabilized carbon) decreased with the C:N ratio. Finally, arenosols are soils for which the adoption of C stocking practices can meet ambitious annual soil carbon storage

objectives. The findings of this study thus generated fresh knowledge on the carbon dynamics of arenosols following a land use change, with a view to application of the 4p1000 Initiative.



## Appendix A

$F^{14}C$ represents the raw radiocarbon activity value and to have access to the age of organic material formed after the 1960
bomb peak. This unit takes isotope fractionation into account and, most importantly, does not depend on the year of
measurement (Reimer et al., 2004) Eq.(A1) :

$$F^{14}C = \frac{A_{SN}}{A_{ON}},\tag{A1}$$

AON is the normalized oxalic acid activity, equivalent to the atmospheric activity in 1950, before the bomb peak. ASN
represents the sample activity, normalised by the isotopic fractionation that occurs via plants during photosynthesis. The
correction is a reference for 13C fractionation with a value of -25‰. As is the sample activity at the dating time Eq.(A2, A3) :

$$A_{SN} = A_S \left( 1 - 2 * \left( \frac{25 + \partial^{13}C}{1000} \right) \right),\tag{A2}$$

$$A_S = \frac{^{14}C \; sample \; atoms}{^{14}C \; sample \; atoms},\tag{A3}$$

However, many authors use Δ14C to express 14C, which represents the deviation of the sample 14C content from the bomb
peak but it depends on the year it was measured, regardless of the age of the sample (Reimer et al., 2004; Stuiver and Polach,
1977). The data expressed in Δ14C is shown Fig.E2 to facilitate comparison with the literature findings, Eq.(A4):

$$\Delta^{14}C = (F^{14}C / e(((years - 1950)/(5730/ln2)) - 1) * 1000,\tag{A4}$$

From the radiocarbon data, it is possible to access a relative age, expressed in years before present (BP). The starting date of
the age scale is 1 January 1950, which was before the bombs peak, and corresponds to the first publications with radiocarbon
dates. The BP age takes the radiocarbon decay equation into account and was calculated according to the Libby half-life of
5568 years (Libby et al., 1949), Eq.(A5):

$$Age \; BP = -5568 * \frac{\ln(F^{14}C)}{ln2},\tag{A5}$$






**Appendix B**

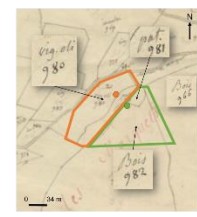

19th century

1808-1848

20th century

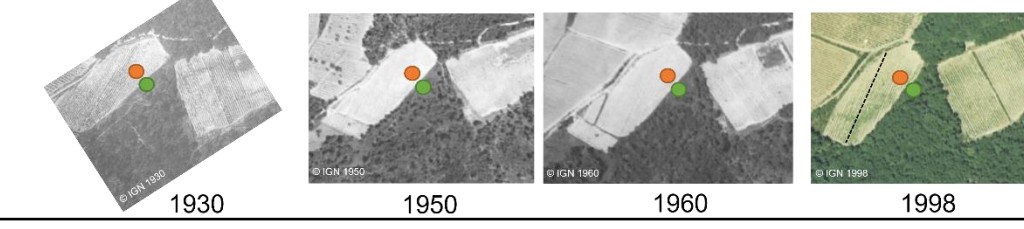

1930       1950       1960       1998

21th century

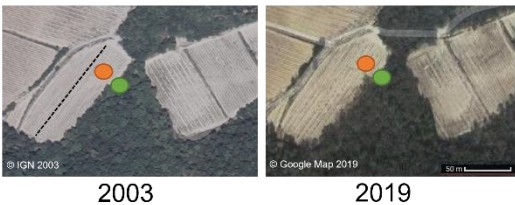

2003       2019

**Figure B1** History of land use at the Les Brugassières site from the 19th century to present day, through the study of old maps and aerial photos. The boundaries of the forest (green) and vineyard (orange) plots were shown on the Napoleonic land register

(1808-1848, https://archives.var.fr). All aerial photos from the 20th century to present day, except 2019, were from the IGN *Remonter le temps* website (https://remonterletemps.ign.fr/, IGN - National Photo Library - [1930, 1950, 1960, 1998, 2003]). The 2019 photo was from © Google Map website (https://maps.google.fr). The pit locations are indicated by orange and green circles.






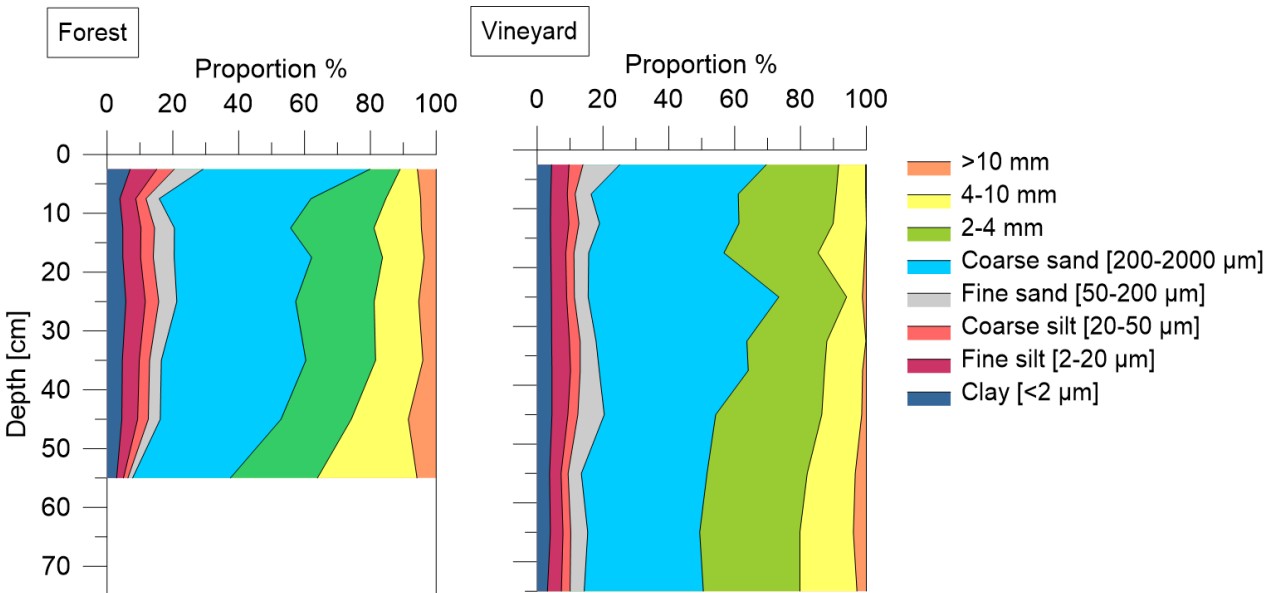

**Figure B2** Granulometry comparison between the pits under the forest and under the vineyard, as a function of depth. The eight grain size fractions, clay <2 µm (dark blue), fine silt 2-20 µm (burgundy), coarse silt 20-50 µm (pink), fine sand 50-200

µm (grey), coarse sand 200-2000 µm (sky blue), 2-4 mm (green), 4-10 mm (yellow) and >10 mm (orange), are expressed in % of the mineral phase. The particle size profiles in the forest and vineyard soils showed about 50% coarse sand and did not vary significantly with depth. There was no significant difference in soil granulometry between the forest plot and the vineyard plot.





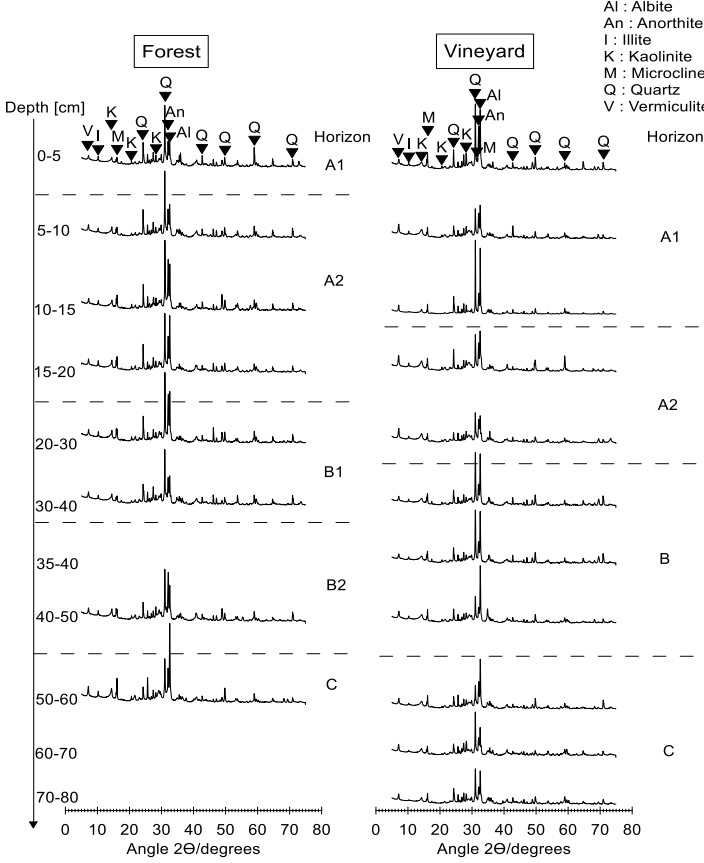

**Figure B3** Forest/vineyard comparison of X-ray diffractograms as a function of soil depth. The rhombs indicate the peak considered and the letters above are the corresponding minerals. The mineralogy was determined by X-ray diffraction on powder samples, deposited on a silicon plate, and measured by a PANalytical X'pert PRO diffractometer, with a cobalt radiation source. The range of 2θ was between 5° and 75°, with a step size of 0.033° and a measurement time of 5 h 10 min per sample. The the forest and cultivated soil mineralogy is characteristic of a granitic bedrock with quartz, feldspar and secondary minerals (illite and vermiculite), throughout the entire profile. The mineralogy was equivalent in both soils.





**Appendix C**

**Table C1** Table of bulk density [g cm$^{-3}$], carbon content [gC kg$^{-1}$], C: N ratio as a function of plant cover and depth. A,B,C are the pit profiles, BD is the bulk density, ± is the analytical error , X is the mean of the 3 profiles A, B and C and SD is the standard deviation.

| | Depth [cm] | BD g cm$^{-3}$ A | B | C | X | SD | TOC gC kg$^{-1}$ A | B | C | X | SD | C:N A | B | C | X | SD |
|---|---|---|---|---|---|---|---|---|---|---|---|---|---|---|---|---|
| **Vineyard** | 0-5 | 1.55 | 1.14 | 1.17 | 1.29 | 0.23 | 2.17 ± 0.27 | 2.38 ± 0.27 | 0.93 ± 0.23 | 1.83 | 0.78 | 7.00 ± 1.57 | 10.80 ± 2.62 | 4.89 ± 1.90 | 7.56 | 3.00 |
| | 5-10 | 1.59 | 0.84 | 1.09 | 1.17 | 0.38 | 1.05 ± 0.23 | 0.79 ± 0.22 | 0.53 ± 0.22 | 0.79 | 0.26 | 8.75 ± 3.71 | | 7.57 ± 5.53 | 8.16 | 0.83 |
| | 10-15 | 1.60 | 0.89 | 0.94 | 1.14 | 0.40 | 1.21 ± 0.24 | 0.61 ± 0.22 | 1.03 ± 0.23 | 0.95 | 0.31 | 10.10 ± 4.02 | | 11.40 ± 5.53 | 10.75 | 0.92 |
| | 15-20 | 1.60 | 0.79 | 0.96 | 1.12 | 0.43 | 0.93 ± 0.23 | 0.73 ± 0.22 | 1.30 ± 0.24 | 0.99 | 0.29 | 10.30 ± 5.21 | 12.20 ± 8.20 | 11.80 ± 4.76 | 11.43 | 1.00 |
| | 20-30 | 0.81 | 1.08 | 1.03 | 0.98 | 0.14 | 0.79 ± 0.22 | 0.67 ± 0.22 | 1.28 ± 0.24 | 0.91 | 0.32 | 9.88 ± 5.63 | | 9.14 ± 3.35 | 9.51 | 0.52 |
| | 30-40 | 1.02 | 0.90 | 0.86 | 0.93 | 0.09 | 0.74 ± 0.22 | 1.35 ± 0.24 | 1.01 ± 0.23 | 1.03 | 0.30 | 10.60 ± 6.59 | 11.50 ± 4.35 | 10.10 ± 4.70 | 10.73 | 0.71 |
| | 40-50 | 1.00 | 0.82 | 0.85 | 0.89 | 0.10 | 1.10 ± 0.23 | 1.42 ± 0.24 | 0.58 ± 0.22 | 1.03 | 0.42 | 12.20 ± 5.76 | | 11.60 ± 9.42 | 11.90 | 0.42 |
| | 50-60 | 1.06 | 0.75 | 1.05 | 0.95 | 0.18 | 0.83 ± 0.23 | 0.94 ± 0.24 | 0.88 ± 0.23 | 0.88 | 0.05 | 11.90 ± 7.04 | | 11.00 ± 5.99 | 11.45 | 0.64 |
| | 60-70 | 0.92 | 0.78 | 0.98 | 0.90 | 0.10 | 0.53 ± 0.22 | 0.69 ± 0.23 | 0.92 ± 0.23 | 0.71 | 0.20 | 8.83 ± 6.87 | | 10.20 ± 5.18 | 9.52 | 0.97 |
| | 70-80 | | | | | | 0.42 ± 0.21 | 0.34 ± 0.22 | 0.87 ± 0.23 | 0.54 | 0.29 | 7.00 ± 6.14 | 11.30 ± 14.98 | 10.90 ± 5.95 | 9.73 | 2.38 |
| **Forest** | 0-5 | 1.21 | 1.42 | | 1.32 | 0.15 | 44.00 ± 1.56 | 32.60 ± 1.21 | 50.7 ± 1.77 | 42.43 | 9.15 | 16.10 ± 1.27 | 2.12 ± 1.27 | 3.13 ± 1.25 | 7.12 | 7.80 |
| | 5-10 | 1.46 | 1.65 | | 1.56 | 0.13 | 23.80 ± 0.94 | 17.50 ± 0.74 | 28.4 ± 1.08 | 23.23 | 5.47 | 14.80 ± 1.30 | 0.00 ± 0.00 | 1.86 ± 1.29 | 5.55 | 8.06 |
| | 10-15 | 1.49 | 1.73 | | 1.61 | 0.17 | 12.50 ± 0.59 | 12.50 ± 0.59 | 15.2 ± 0.67 | 13.40 | 1.56 | 13.50 ± 1.42 | 0.88 ± 1.50 | 1.08 ± 1.39 | 5.15 | 7.23 |
| | 15-20 | 1.48 | 1.40 | | 1.44 | 0.06 | 7.72 ± 0.44 | 9.29 ± 0.49 | 10.5 ± 0.53 | 9.17 | 1.39 | 13.30 ± 1.70 | 0.00 ± 0.00 | 0.78 ± 1.50 | 4.69 | 7.46 |
| | 20-30 | 1.65 | 1.62 | | 1.63 | 0.02 | 5.18 ± 0.36 | 5.94 ± 0.38 | 7.87 ± 0.44 | 6.33 | 1.39 | 12.00 ± 1.83 | 0.48 ± 1.76 | 0.59 ± 1.68 | 4.36 | 6.62 |
| | 30-40 | 1.74 | 1.67 | | 1.70 | 0.05 | 4.34 ± 0.33 | 4.16 ± 0.33 | 4.98 ± 0.35 | 4.49 | 0.43 | 11.70 ± 1.96 | 0.36 ± 1.97 | 0.39 ± 2.02 | 4.15 | 6.54 |
| | 40-50 | 1.72 | 1.55 | | 1.64 | 0.12 | 3.44 ± 0.31 | 3.41 ± 0.31 | 3.32 ± 0.30 | 3.39 | 0.06 | 10.80 ± 2.02 | 0.32 ± 2.00 | 0.30 ± 2.15 | 3.81 | 6.06 |
| | 50-60 | 1.70 | 1.58 | | 1.64 | 0.09 | 2.70 ± 0.28 | 1.89 ± 0.26 | 2.25 ± 0.27 | 2.28 | 0.41 | 10.40 ± 2.26 | 0.18 ± 2.98 | 0.19 ± 3.09 | 3.59 | 5.90 |
| | 60-70 | 1.74 | 1.68 | | 1.71 | 0.04 | 2.09 ± 0.26 | | | 2.09 | | 9.95 ± 2.57 | | | 9.95 | |


**Table C2** F$^{14}$C per composite sample (A+B+C) or per profile (A, B and C) according to soil depth, the numbers 1, 2 or 3 indicate the repetitions, ± is the analytical error.

| | Depth [cm] | A+B+C C1 [µg] | F$^{14}$C 1 | C2 [µg] | F$^{14}$C 2 | C3 [µg] | F$^{14}$C 3 | A C1 [µg] | F$^{14}$C 1 | C2 [µg] | F$^{14}$C 2 | C3 [µg] | F$^{14}$C 3 | B C1 [µg] | F$^{14}$C 1 | C2 [µg] | F$^{14}$C 2 | C C1 [µg] | F$^{14}$C 1 | C2 [µg] | F$^{14}$C 2 |
|---|---|---|---|---|---|---|---|---|---|---|---|---|---|---|---|---|---|---|---|---|---|
| **Vineyard** | 0-5 | 71.3 | 0.990 ± 0.009 | | | | | | | | | | | | | | | | | | |
| | 5-10 | 59.3 | 0.937 ± 0.009 | 54.8 | 0.933 ± 0.009 | 54.0 | 0.933 ± 0.009 | 54.8 | 0.969 ± 0.009 | | | | | 54.8 | 0.926 ± 0.009 | | | 26.3 | 0.897 ± 0.015 | 27.8 | 0.880 ± 0.013 |
| | 10-15 | 54.0 | 0.984 ± 0.009 | | | | | | | | | | | | | | | | | | |
| | 15-20 | 45.0 | 0.968 ± 0.010 | | | | | | | | | | | | | | | | | | |
| | 20-30 | 50.3 | 0.977 ± 0.010 | | | | | | | | | | | | | | | | | | |
| | 30-40 | 55.5 | 0.989 ± 0.009 | | | | | | | | | | | | | | | | | | |
| | 40-50 | 50.3 | 1.014 ± 0.010 | 53.3 | 1.008 ± 0.009 | 53.3 | 1.024 ± 0.009 | 48.0 | 0.967 ± 0.009 | | | | | 86.3 | 1.081 ± 0.009 | | | 27.0 | 0.909 ± 0.013 | | |
| | 50-60 | 51.4 | 0.966 ± 0.009 | | | | | 40.0 | 0.981 ± 0.009 | | | | | 36.0 | 0.977 ± 0.009 | | | 34.0 | 0.959 ± 0.009 | | |
| | 60-70 | 25.7 | 0.959 ± 0.014 | | | | | | | | | | | | | | | | | | |
| | 70-80 | 24.0 | 0.893 ± 0.014 | | | | | | | | | | | | | | | | | | |
| **Forest** | 0-5 | 998.0 | 1.089 ± 0.003 | | | | | | | | | | | | | | | | | | |
| | 5-10 | 1 000.0 | 1.118 ± 0.002 | 988.0 | 1.116 ± 0.002 | | | 1 000.0 | 1.113 ± 0.003 | 987.0 | 1.118 ± 0.003 | 123.8 | 1.122 ± 0.009 | 998.0 | 1.095 ± 0.002 | 108.0 | 1.124 ± 0.009 | 997.0 | 1.115 ± 0.003 | 93.8 | 1.120 ± 0.009 |
| | 10-15 | 997.0 | 1.108 ± 0.002 | | | | | | | | | | | | | | | | | | |
| | 15-20 | 996.0 | 1.104 ± 0.002 | | | | | | | | | | | | | | | | | | |
| | 20-30 | 983.0 | 1.059 ± 0.002 | | | | | | | | | | | | | | | | | | |
| | 30-40 | 986.0 | 1.015 ± 0.002 | | | | | | | | | | | | | | | | | | |
| | 40-50 | 86.6 | 0.995 ± 0.009 | 86.57 | 0.980 ± 0.009 | 86.57 | 0.980 ± 0.009 | 90.9 | 0.991 ± 0.008 | | | | | 90.9 | 0.974 ± 0.008 | | | 86.6 | 1.005 ± 0.009 | | |
| | 50-60 | 91.7 | 0.964 ± 0.008 | | | | | | | | | | | | | | | | | | |
| | 60-70 | 89.1 | 0.986 ± 0.008 | | | | | | | | | | | | | | | | | | |



## Appendix D

**Table D1** listing papers used in this study, with land use type for each soil and associated TOC as a function of depth. These papers all deal with arenosols, or at least sandy soils, in Mediterranean climates according to the Köppen-Geiger criteria. They were found by accessing the ISRaD database or the Web of Science with the keywords "$^{14}$C arenosol heterogeneity".

| Paper | Publication year | Country | DOI | Soil type/major texture | Climat | Land use | Plot Age years | Depth cm | TOC g kg$^{-1}$ | ET g kg$^{-1}$ |
|---|---|---|---|---|---|---|---|---|---|---|
| Andreetta et al. | 2013 | Italy | 10.1007/s10533-011-9693-9 | Haplic Arenosol | csa | Holm oak forest | 50 | 0-5 | 104.7 | na |
| Andreetta et al. | 2013 | Italy | 10.1007/s10533-011-9693-9 | Haplic Arenosol | csa | Holm oak forest | 50 | 5-11 | 9.3 | na |
| Andreetta et al. | 2013 | Italy | 10.1007/s10533-011-9693-9 | Haplic Arenosol | csa | Holm oak forest | 50 | 11-30 | 14.2 | na |
| Andreetta et al. | 2013 | Italy | 10.1007/s10533-011-9693-9 | Haplic Arenosol | csa | Holm oak forest | 50 | 30-55 | 6.1 | na |
| Andreetta et al. | 2013 | Italy | 10.1007/s10533-011-9693-9 | Haplic Arenosol | csa | Holm oak forest | 50 | 55-75 | 2.1 | na |
| Andreetta et al. | 2013 | Italy | 10.1007/s10533-011-9693-9 | Haplic Arenosol | csa | Holm oak forest | 50 | 75-120 | 1.4 | na |
| Caravaca et al. | 2002 | Spain | 10.1016/S0167-1987(02)00080-6 | Calcaric Arenosol | csa | Spontaneous grass cover | na | 0-20 | 21.3 | na |
| Caravaca et al. | 2002 | Spain | 10.1016/S0167-1987(02)00080-6 | Calcaric Arenosol | csa | vineyard | na | 0-20 | 3.2 | na |
| Conradie | 2001 | South Africa | https://doi.org/10.21548/22-2-2192 | Sandy soil | scb | vineyard | na | 0-20 | 4.8 | na |
| Conradie | 2001 | South Africa | https://doi.org/10.21548/22-2-2192 | Sandy soil | scb | vineyard | na | 20-40 | 1.7 | na |
| Conradie | 2001 | South Africa | https://doi.org/10.21548/22-2-2192 | Sandy soil | scb | vineyard | na | 40-60 | 1.6 | na |
| Fierro et al. | 2007 | Italy | 10.1071/WF06114 | Calcaric Arenosol | csa | forest | na | 0-5 | 47 | 7 |
| Fierro et al. | 2007 | Italy | 10.1071/WF06114 | Calcaric Arenosol | csa | forest | na | 0-5 | 48 | 8 |
| Fierro et al. | 2007 | Italy | 10.1071/WF06114 | Calcaric Arenosol | csa | forest | na | 0-5 | 45 | 15 |
| Fierro et al. | 2007 | Italy | 10.1071/WF06114 | Calcaric Arenosol | csa | forest | na | 0-5 | 50 | 21 |
| Fierro et al. | 2007 | Italy | 10.1071/WF06114 | Calcaric Arenosol | csa | forest | na | 0-5 | 54 | 21 |
| Fierro et al. | 2007 | Italy | 10.1071/WF06114 | Calcaric Arenosol | csa | forest | na | 0-5 | 48.8 | 14.4 |
| Fourie et al. | 2005 | South Africa | https://doi.org/10.21548/26-2-2129 | Sandy soil | csb | vineyard | na | 0-30 | 1.3 | na |
| Fourie et al. | 2005 | South Africa | https://doi.org/10.21548/26-2-2129 | Sandy soil | csb | vineyard | na | 30-60 | 1.0 | na |
| López-Piñeiro et al. | 2013 | Spain | http://dx.doi.org/10.1016/j.still.2012.09.007 | Loamy sand soil | csa | vineyard | na | 0-10 | 1.73 | na |
| López-Piñeiro et al. | 2013 | Spain | http://dx.doi.org/10.1016/j.still.2012.09.007 | Loamy sand soil | csa | vineyard | na | 0-10 | 1.68 | na |
| Nogales et al. | 2018 | Portugal | 10.3389/fpls.2018.01906 | Arenosol | csa | vineyard | na | 0-30 | 5.7 | 0.213 |
| Okur et al. | 2009 | Turkey | 10.3906/tar-0806-23 | Sandy loamy soil | csa | vineyard | na | 0-20 | 7.8 | na |
| Pinzari et al. | 1999 | Italy | 10.1016/S0167-7012(99)00007-X | Sandy soil | csa | Natural oak foret | 100 | 0-20 | 16 | 1.62 |
| Pinzari et al. | 1999 | Italy | 10.1016/S0167-7012(99)00007-X | Sandy soil | csa | Natural oak foret | 100 | 20-40 | 5.6 | 0.33 |
| Pinzari et al. | 1999 | Italy | 10.1016/S0167-7012(99)00007-X | Sandy soil | csa | maquis | na | 0-20 | 31 | 2.45 |
| Pinzari et al. | 1999 | Italy | 10.1016/S0167-7012(99)00007-X | Sandy soil | csa | maquis | na | 20-40 | 7.7 | 0.45 |
| Pinzari et al. | 1999 | Italy | 10.1016/S0167-7012(99)00007-X | Sandy soil | csa | pine forest plantation | 60 | 0-20 | 20.1 | 2.56 |
| Pinzari et al. | 1999 | Italy | 10.1016/S0167-7012(99)00007-X | Sandy soil | csa | pine forest plantation | 60 | 20-40 | 6.1 | 0.25 |
| Pinzari et al. | 1999 | Italy | 10.1016/S0167-7012(99)00007-X | Sandy soil | csa | natural mixed forest | na | 0-20 | 22.7 | 2.06 |
| Pinzari et al. | 1999 | Italy | 10.1016/S0167-7012(99)00007-X | Sandy soil | csa | natural mixed forest | na | 20-40 | 19 | 0.41 |
| Vittori Antisari et al. | 2016 | Italy | 10.1007/s12665-016-5581-x | Haplic arenosol | csa | Holm forest | na | 0-3 | 72.0 | 5.6 |
| Vittori Antisari et al. | 2016 | Italy | 10.1007/s12665-016-5581-x | Haplic arenosol | csa | Holm forest | na | 3-7 | 49.3 | 2.4 |
| Vittori Antisari et al. | 2016 | Italy | 10.1007/s12665-016-5581-x | Haplic arenosol | csa | Holm forest | na | 7-12 | 10.5 | 0.5 |
| Vittori Antisari et al. | 2016 | Italy | 10.1007/s12665-016-5581-x | Haplic arenosol | csa | Holm forest | na | 12-50 | 1.8 | 0.3 |
| Vittori Antisari et al. | 2016 | Italy | 10.1007/s12665-016-5581-x | Haplic arenosol | csa | Pine forest | na | 0-3 | 42.7 | 1.5 |
| Vittori Antisari et al. | 2016 | Italy | 10.1007/s12665-016-5581-x | Haplic arenosol | csa | Pine forest | na | 3-11 | 10.5 | 1.3 |
| Vittori Antisari et al. | 2016 | Italy | 10.1007/s12665-016-5581-x | Haplic arenosol | csa | Pine forest | na | 11-25 | 1.9 | 0.4 |
| Vittori Antisari et al. | 2016 | Italy | 10.1007/s12665-016-5581-x | Haplic arenosol | csa | Pine forest | na | 25-50 | 1.2 | 0.3 |
| Vittori Antisari et al. | 2016 | Italy | 10.1007/s12665-016-5581-x | Brunic arenosol | csa | Hygro forest (oak) | 245 | 0-3 | 49.7 | 3.1 |
| Vittori Antisari et al. | 2016 | Italy | 10.1007/s12665-016-5581-x | Brunic arenosol | csa | Hygro forest | 245 | 3-6 | 19.6 | 1.2 |
| Vittori Antisari et al. | 2016 | Italy | 10.1007/s12665-016-5581-x | Brunic arenosol | csa | Hygro forest | 245 | 6-12 | 9.6 | 1.8 |
| Vittori Antisari et al. | 2016 | Italy | 10.1007/s12665-016-5581-x | Brunic arenosol | csa | Hygro forest | 245 | 12-19 | 2.1 | 0.3 |
| Vittori Antisari et al. | 2016 | Italy | 10.1007/s12665-016-5581-x | Brunic arenosol | csa | Hygro forest | 245 | 19-30 | 2.1 | 0.6 |
| Vittori Antisari et al. | 2016 | Italy | 10.1007/s12665-016-5581-x | Brunic arenosol | csa | Hygro forest | 245 | 30-50 | 10.0 | 0.2 |





**Table D2** Köppen-Geiger criteria. The Köppen-Geiger Mediterranean climate classes including the defining criteria, adapted
from (Beck et al., 2018): MAT = mean annual air temperature (°C); Tcold = air temperature of the coldest month (°C); Thot
= air temperature of the warmest month (°C); Tmon10 = the number of months with air temperature >10 °C (unitless); MAP
= mean annual precipitation (mm y−1); Psdry = precipitation in the driest month in summer (mm month−1); Pwdry =
precipitation in the driest month in winter (mm month−1); Pswet = precipitation in the wettest month in summer (mm
month−1); Pwwet = precipitation in the wettest month in winter (mm month−1); Pthreshold =2×MAT if >70% of precipitation
falls in winter, Pthreshold =2×MAT+28 if >70% of precipitation falls in summer, otherwise Pthreshold =2×MAT+14. Summer
(winter) is the 6-month period that is warmer (colder) between April-September and October-March.

| 1st | 2nd | 3rd | Description | Criterion |
|-----|-----|-----|-------------|-----------|
| B | | | Arid | $MAP < 10 \times P_{threshold}$ |
| | W | | desert | $MAP < 5 \times P_{threshold}$ |
| | S | | steppe | $MAP \geq 5 \times P_{threshold}$ |
| | | h | hot | $MAT \geq 18$ |
| | | k | cold | $MAT < 18$ |
| C | | | Temperate | Not (B) & $T_{hot} > 10$ & $0 < T_{cold} < 18$ |
| | s | | Dry summer | $P_{sdry} < 40$ & $P_{sdry} < P_{wwet}/3$ |
| | f | | without dry season | Not (Cs) |
| | | a | hot summer | $T_{hot} \geq 22$ |
| | | b | warm summer | Not (a) & $T_{mon10} \geq 4$ |
| | | c | cold summer | Not (a or b) & $1 \leq T_{mon10} < 4$ |





**Appendix E**

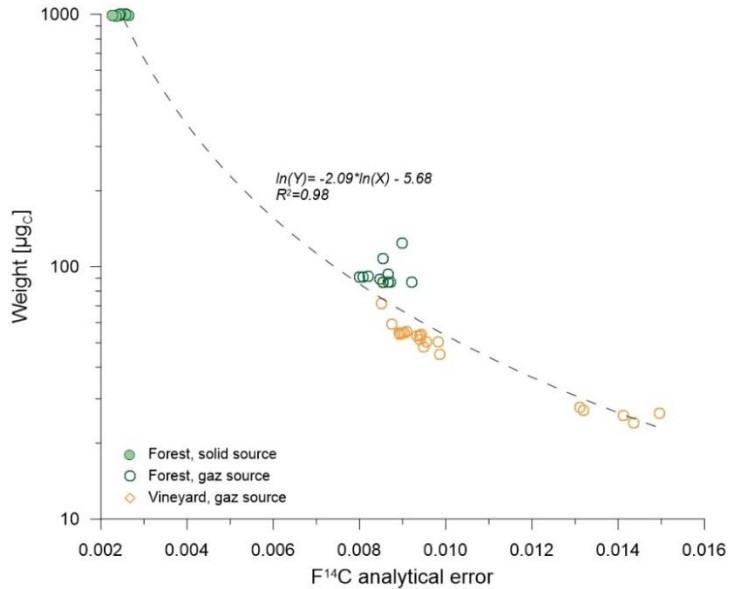


**Figure E1** Influence of the carbon mass of the measured sample on the analytical error of ECHoMICADAS. The solid green circles represent soil samples obtained under the forest analyzed with the solid source, the empty green circles those analyzed with the gas source and the empty orange triangles are the soil samples obtained under vines analyzed with the gas source.





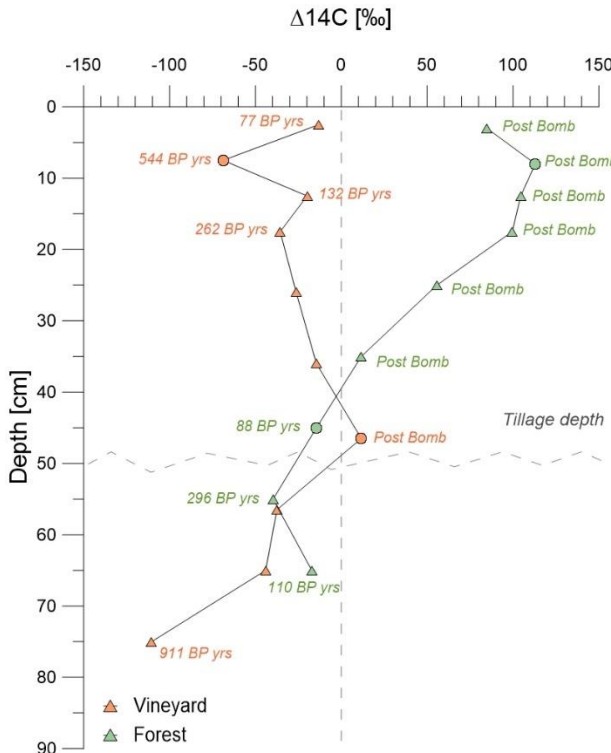

**Figure E2** Comparison of the BP age patterns, via $\Delta^{14}C$ [‰], as a function of the soil depth and vegetation cover.



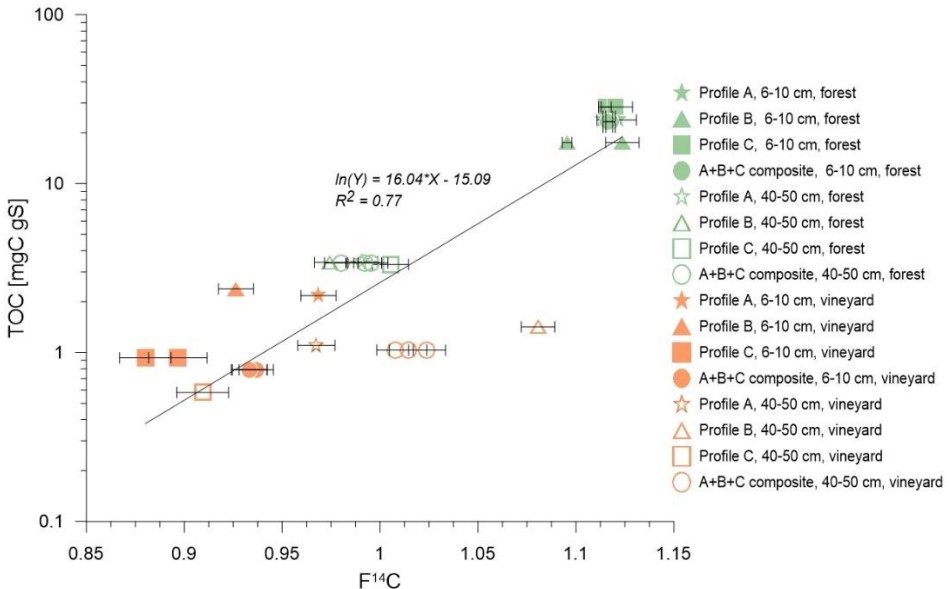

**Figure E3** Variations in carbon content as a function of $F^{14}C$. Profiles A, B and C are represented by stars, triangles and squares, respectively. These symbols are solid when they represent surface samples (6-10 cm) and empty when they represent

deep samples (40-50 cm). Soil samples obtained under the forest are green and those under vines are orange. The error bars represent the analytical error. The TOC values were higher with younger $F^{14}C$ (usually topsoil samples): $R^2=0.77$. Under the vineyard, ploughing had eliminated the young carbon pools.

**Author contributions**

The conceptualization of the study for this paper was done by SQ, CH and IBD with input from NC, FJ, DB, AD and SC. All
the authors participated in the collection of resources. The investigation was done by SQ, IBD and CH with substantial input from FJ, NC, AD, SC and DB. The data curation and formal analysis and methodology were done by SQ, IBD and CH. The visualization for the paper was performed by SQ, with substantial input from IBD and CH as well as feedback from all authors. SQ and IBD wrote the initial draft and all authors were involved in the review and editing of the paper.

**Competing interests**

The authors declare that they have no conflict of interest.



**Acknowledgements**

We would like to thank Jérôme Balesdent, who sadly left us too soon, for his precious support. We would also like to thank Mr. Février and Mr. Coulomb, president of the Vignerons du Plan de la Tour cooperative, for allowing us to sample soils in their plots. Finally, we would like to thank Frédéric Guibal for his dendrochronology expertise.

**Financial support**

This research was funded by the ANR (NanoSoilC project ANR-16-CE01-0012-02) and supported by the University of Aix-Marseille. We also thank CNRS INSU for additional financial support to Solène Quero (PhD) in the context of the Covid-19 pandemic.

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
