# Peer review of "Dynamics of carbon loss from an arenosol by a forest/vineyard land use change on a centennial scale"

_SOIL, 2021_

## Author Comment (AC1)

**Comment on soil-2021-115**

**Anonymous Referee #1**
**Referee comment on "Dynamics of carbon loss from an arenosol by a forest/vineyard landuse change on a centennial scale" by Solène Quéro et al., SOIL Discuss.**

The present study wants to give insights about the effect of the conversion of forest to vineyard grown on organic carbon dynamics in arenosol. The manuscript is not acceptable for publication because of the poor quality of the text and for the lack of true replicates.

"poor quality of the text":
=> Since the reviewer has not provided any elements illustrating the poor quality of the text, it was hard for us to improve it, notably as: (1) from the grammar and syntax standpoint, the English was corrected by a professional English-speaking scientific translator, and (2) Referee #2 confirmed that the English was correct and did not highlight any issues with regard to the linguistic quality of the text. Upon re-reading, however, we recognize that some of the wording may have been a bit clumsy and that some arguments could have been better presented. We will try to improve the text as much as possible. "

"lack of true replicates"
=> Studies that only aim at mapping stocks or studying stock variations at regional or even plot scales justify the multiplication of sampling points ("mosaic strategy" of Eldon and Gershenson, 2015). Our objective was different—we sought to compare the impact of a change in land-use and associated agricultural practices on carbon dynamics at the scale of a soil profile (including topsoil and subsoil, down to the parent rock). We therefore opted for a profile comparison approach (paired-site strategy as defined by Eldon and Gershenson, 2015), as for example Torn et al. (1997) did with 14C in their article published in Nature, and as Jagercikova et al. (2015) did for 10Be. These cosmogenic isotope approaches allow access to the soil processes, contrary to simple stock analysis, yet these are very high cost analyses so only a small number of samples can be screened (Mathieu et al., 2015). This last aspect is generally offset by careful selection of the profile analysed, thereby ensuring the comparability (see description below). The study of Laurence et al. (2020)—where a large number of paired-sites were investigated with 14C approaches—illustrates the interest of this type of approach in soil carbon dynamics.

For this reason, we conducted a detailed analysis at the pit scale. By using $^{14}$C, we had two objectives: (1) to compare the dynamics of carbon (from topsoil to parent rock) in crop and forest soils, and (2) to determine the $^{14}$C variability at the pit scale. Our results showed that a composite sample was highly representative of the mean $^{14}$C trend. Otherwise, had we not studied the variability on the different sides of the pits, we would not have been able to demonstrate the very marked differences between the two sites: in the forest, there was very low variance at a single depth, whereas this variance was very high above 50 cm in the vineyard. This finding highlighted the effect of agricultural practices (deep ploughing) on the C dynamics. Without this fine analysis, we would not have been able to reach this conclusion. This warrants our sampling protocol.

However, we would like to point out to Referee #1 that the location of these pits was based on a careful choice of the sampled profile, i.e. they only differed in terms of land use. This selection was carried out in 7 successive stages:

1. In the French Mediterranean area, a granitic pluton outcrop was sought to make sure that arenosols would be present: the granite of Plan de la Tour (Maures, South of France, represented by the north-south elongated red zone, in the center

of the geological map below (source https://www.geoportail.gouv.fr/carte) (Figure 1).

[Figure]

*Figure 1 : geological map (source https://www.geoportail.gouv.fr/carte)*

2. In the Plan-de-la-Tour granite area, places with adjacent vineyard and forest plots were identified on the basis of satellite images.

3. To be sure that the forest C dynamics were representative of a forest pedogenesis and not the result of recent afforestation, we selected only sites already in forest in the 1800s ( Napoleonic land register  1808-1848, see Supplementary information, and Ordnance Survey map, 1820-1866, see Figure 2).

4. Among these sites, we selected only those with comparable topographic situation for two land uses and ideally with the flattest possible topography in so as to minimize differences in C dynamics that could result from differential erosion between crop fields and forests. (topographic map at 1:25,000 scale)

5. We went to the fields at the 5 sites selected based on the above criteria. We then selected one site ("Les Brugassières") according to their accessibility and the sampling authorizations.

6. The location of the sampling pit was chosen on the basis of a structural analysis, as is conventionally done in pedology studies (e.g.Humbel, 1987). We then augered to identify points where the soil was: (1) sufficiently deep (about 80 cm), (2) equivalent depths in the forest and crop soils, and (3) where there was very little distance between 80 cm-deep crop and forest soils (less than 20 m). We sought to find an area of the plot where the two pit sites would likely have identical pedogenesis patterns prior to vine planting.

7. Finally, we performed a screening (0-30 cm topsoil layer) to assess the homogeneity of total organic carbon contents in vineyard plots and adjacent forests.

Following these successive eliminations, the precise pit locations were chosen (Figure 2) :

| Topography | Pedologic map | Geologic map | Ordnance Survey map |

[Figure]

*Figure 2 : topographic , pedologic (Source: Soil reference system of the VAR), geological and survey map (source :https://www.geoportail.gouv.fr).Brown arrow = crop; green arrow = forest.*

This selection process seemed relevant to compare the evolution patterns of a soil associated with cultivation and agricultural practices in a detailed way and at the profile scale. We will outline this methodology in the revised version.

The introduction should be deeply revised. It includes several information written without following a clear framework.
=> It would be hard to precisely address the concerns of Referee #1 here because it is unclear what information actually needs to be better documented in the framework and exactly why the introduction needs deep revision. We improved it according to the recommendations of Referee #2 and as described below.

The aims seem to be not linked to the state of the art stated in the introduction section (e.g., you did not introduce the effect of vineyard on soil)
=> We agree with the Referee regarding the specific effects of the vineyard that were not detailed in the introduction. We will do so. This will help us reorganize the introduction as recommended. We propose to add the following paragraph:

*"Relative to arable and pasture systems, SOC studies in vineyards have received less attention" (Payen et al., 2021), while viticulture is now a major agricultural growth sector under Mediterranean climatic conditions worldwide (Eldon and Gershenson, 2015). Yet, at the same time, vineyards in Mediterranean regions are among the most degraded agricultural crop systems (Ferreira et al., 2020). In their metanalysis of cultivated-uncultivated Mediterranean paired sites, Eldon and Gershenson, (2015) found that the soil carbon storage loss ranged from -30.6% to -52.1%, with the highest losses noted with the conversion to vineyards. Land degradation in Mediterranean vineyards is associated with loss of soil organic matter due to accelerated mineralization, decreased nutrient content, topsoil compaction and reduced water infiltration capacity, enhanced soil erosion rates, accumulation of metals and organic pollutants, and associated loss of soil biodiversity due to habitat deterioration (Bogunovic et al., 2019; Bordoni et al., 2019; Eldon and Gershenson, 2015; Ferreira et al., 2020, 2018; Kratschmer et al., 2018). These degradations result from a combination of relatively poor soils that prevail in Mediterranean regions, time of plantation and parent material, steep slopes, intense rainfall and, overall, the intensive crop management practices. These traditional wine-growing practices involve frequent tillage to minimize weed cover and soil compaction, postharvest removal of crop residues, and high mineral fertilizer and phytopharmaceutical compound application rates (Ferreira et al., 2020)"*

Otherwise, we believe that the points outlined in the introduction showcase the state of the art and justify the approaches we implemented.

Ls 26-27: what do you mean for "and occur in layers about 100 cm deep"?
=> We agree that this sentence was not clear and propose to change the beginning of the introduction as follows:

*"Arenosol is one of the 30 soil groups in the FAO soil classification system. Arenosols account for about 7% of the world's soils and are found mostly under desert, tropical and Mediterranean climatic conditions. They are silty-sandy or sandy soils, with less than 35% by volume of coarse elements, exhibit no or partial diagnostic horizon and are generally 100 cm deep. Given their excessive permeability and low nutrient content, agricultural use of arenosols requires careful management."*

Ls 27-28: remove "for the richest" and "for the poorest"
=> We agree to remove these qualifiers that do not provide additional information.

L 27: what do you mean for "surface"
=> We will replace "surface" by "topsoil" here and elsewhere in the paper.

Ls 30-31: you could write "the conversion from forest or grassland to cropland can lead to......
=>Yes, we will change the sentence as proposed: *"As with other soil types, the conversion form forest or grassland to cropland can lead to a loss of carbon (Lal, 2004).*

Ls 31-32: all soil types are suitable to store carbon and to meet the 4 per 1000 initiative
=>Yes, we agree that all types of soil are suitable for carbon storage. We did not mean that only arenosols are suitable, but Referee #1 is right, the sequence of the sentences might suggest this.
In order to follow the logic of the paragraph on the effects of the forest/crop conversion, we will move this sentence to a paragraph dedicated to C storage later in the introduction. This will also be in line with the request of Referee #1 to reorganize the introduction.

L 36: the organic carbon loss, always occur after the conversion of forest and grassland to cropland, therefore the brackets should be removed and the sentence rephrased
=>OK, we will rephrase the sentence as follows: "*Loss of carbon due to conversion from forest or grassland to cropland is linked to the acceleration of erosion, runoff and/or mineralisation and could lead to a C depletion rate of about 50% in 10 years*".

L 37: remove "(CO2 release)"
=> Ok, see sentence above.

L 43: where?
=>We do not really understand the Referee's question here. All agricultural research institutes run experimental sites. In France, the Grignon experimental site, for example, has been devoted to agronomy research since 1826.

L 49: "above criteria", which one?
We mean the criteria mentioned on line 47, namely "*same soil, same climatic conditions, same bedrock, flat topography*". We will clarify the sentence.

The quality of the materials and methods section is poor
=>We will rewrite the material and methods section. We will also note the statistical methods used (as requested Referee #2).

The present study does not have true replicates, it has just subreplicates. In order to satisfy the purposes of the present study, at least 3 soil profiles must be dug in each plot otherwise the findings cannot be considered valuable.

=>See our response at the beginning concerning the paired-site strategy. Following the advice of Referee #2, we also strengthened our results by using statistical approaches to confirm the observed differences between crop and forest soils.

**References**

Bogunovic, I., Pereira, P., Kisic, I., Birkás, M., Rodrigo-Comino, J., 2019. Spatiotemporal Variation of Soil Compaction by Tractor Traffic Passes in a Croatian Vineyard 12.

Bordoni, M., Vercesi, A., Maerker, M., Ganimede, C., Reguzzi, M.C., Capelli, E., Wei, X., Mazzoni, E., Simoni, S., Gagnarli, E., Meisina, C., 2019. Effects of vineyard soil management on the characteristics of soils and roots in the lower Oltrepò Apennines (Lombardy, Italy). Science of The Total Environment 693, 133390. https://doi.org/10.1016/j.scitotenv.2019.07.196

Eldon, J., Gershenson, A., 2015. Effects of Cultivation and Alternative Vineyard Management Practices on Soil Carbon Storage in Diverse Mediterranean Landscapes: A Review of the Literature 37.

Ferreira, C.S., Veiga, A., Caetano, A., Gonzalez-Pelayo, O., Karine-Boulet, A., Abrantes, N., Keizer, J., Ferreira, A.J., 2020. Assessment of the Impact of Distinct Vineyard Management Practices on Soil Physico-Chemical Properties. Air, Soil and Water Research 13, 117862212094484. https://doi.org/10.1177/1178622120944847

Ferreira, C.S.S., Keizer, J.J., Santos, L.M.B., Serpa, D., Silva, V., Cerqueira, M., Ferreira, A.J.D., Abrantes, N., 2018. Runoff, sediment and nutrient exports from a Mediterranean vineyard under integrated production: An experiment at plot scale. Agriculture, Ecosystems & Environment 256, 184–193. https://doi.org/10.1016/j.agee.2018.01.015

Humbel, F.X., 1987. STRUCTURAL ANALYSIS OF SOIL MANTLES AND ORIENTATED DESIGNS OF AGRONOMIC EXPERIMENTS. Land developement - Mangement of acid soils 153–162.

Jagercikova, M., Cornu, S., Bourlès, D., Antoine, P., Mayor, M., Guillou, V., 2015. Understanding long-term soil processes using meteoric 10Be: A first attempt on loessic deposits. Quaternary Geochronology 27, 11–21. https://doi.org/10.1016/j.quageo.2014.12.003

Jiang, Y., Luo, C., Zhang, D., Ostle, N.J., Cheng, Z., Ding, P., Shen, C., Zhang, G., 2020. Radiocarbon evidence of the impact of forest-to-plantation conversion on soil organic carbon dynamics on a tropical island. Geoderma 375, 114484. https://doi.org/10.1016/j.geoderma.2020.114484

Kratschmer, S., Pachinger, B., Schwantzer, M., Paredes, D., Guernion, M., Burel, F., Nicolai, A., Strauss, P., Bauer, T., Kriechbaum, M., Zaller, J.G., Winter, S., 2018. Tillage intensity or landscape features: What matters most for wild bee diversity in vineyards? Agriculture, Ecosystems & Environment 266, 142–152. https://doi.org/10.1016/j.agee.2018.07.018

Lal, R., 2004. Soil Carbon Sequestration Impacts on Global Climate Change and Food Security. Science 304, 1623–1627. https://doi.org/10.1126/science.1097396

Lawrence, C.R., Beem-Miller, J., Hoyt, A.M., Monroe, G., Sierra, C.A., Stoner, S., Heckman, K., Blankinship, J.C., Crow, S.E., McNicol, G., Trumbore, S., Levine, P.A., Vindušková, O., Todd-Brown, K., Rasmussen, C., Hicks Pries, C.E., Schädel, C., McFarlane, K., Doetterl, S., Hatté, C., He, Y., Treat, C., Harden, J.W., Torn, M.S., Estop-Aragonés, C., Asefaw Berhe, A., Keiluweit, M., Della Rosa Kuhnen, Á., Marin-Spiotta, E., Plante, A.F., Thompson, A., Shi, Z., Schimel, J.P., Vaughn, L.J.S., von Fromm, S.F., Wagai, R., 2020. An open-source database for the synthesis of soil radiocarbon data: International Soil Radiocarbon Database (ISRaD) version 1.0. Earth Syst. Sci. Data 12, 61–76. https://doi.org/10.5194/essd-12-61-2020

Mathieu, J.A., Hatté, C., Balesdent, J., Parent, É., 2015. Deep soil carbon dynamics are driven more by soil type than by climate: a worldwide meta-analysis of radiocarbon profiles. Glob Change Biol 21, 4278–4292. https://doi.org/10.1111/gcb.13012

Payen, F.T., Sykes, A., Aitkenhead, M., Alexander, P., Moran, D., MacLeod, M., 2021. Soil organic carbon sequestration rates in vineyard agroecosystems under different soil management practices: A meta-analysis. Journal of Cleaner Production 13. https://doi.org/10.1016/j.jclepro.2020.125736

Torn, M.S., Trumbore, S.E., Chadwick, O.A., Vitousek, P.M., Hendricks, D.M., 1997. Mineral control of soil organic carbon storage and turnover. Nature 389, 170–173. https://doi.org/10.1038/38260

---

## Author Comment (AC2)

**Comment on soil-2021-115**

Anonymous Referee #2
Referee comment on "Dynamics of carbon loss from an arenosol by a forest/vineyard land use change on a centennial scale" by Solène Quéro et al., SOIL Discuss., https://doi.org/10.5194/soil-2021-115-RC2, 2022

The manuscript "Dynamics of carbon loss from an arenosol by a forest/vineyard land use change on a centennial scale" presents, as tittle says, the results of a research about long term variations in soil organic carbon (SOC) stocks and their dynamics in a 80 cm deep Mediterranean Arenosol that had undergone a land use change from forest to vineyard over more than 100 years. According to their results a stock of 50 GtC ha-1 in the 0-30 cm forest soil horizon was reduced to 3 GtC ha-1 after long-term grape cultivation. Analyses of 14C showed that deep ploughing (50 cm) in vineyard plot redistributed the remaining carbon both vertically and horizontally. Authors concluded that this soil would have a high carbon storage potential if agricultural practices, such as grassing or organic amendment applications, were to be implemented within the framework of the 4 per 1000 Initiative.

The text denotes a considerable amount of field and laboratory work. In general, manuscript is well written (English grammar and spelling are correct). It is a very interesting research dealing with SOC stocks in soil profiles under different land uses. The natural radiocarbon (14C) abundance analyses present a significant contribution to the discussion. References are updated and they support properly introduction and discussion sections. Tables and Figures are of good quality and all necessary. However, I consider manuscript needs a MODERATE revision before being accepted for publication. It needs to consider the following remarks.

=>We thank Referee #2 for this positive evaluation and for all of the suggestions proposed to improve the paper.

**General comments:**

There are not statistical analyses supporting the data discussion. Authors are comparing values and treatments and this should be done by means of statistics.

=>We have taken this important remark of Referee into account and have called upon the expertise of a statistician (Joel Chadoeuf) with whom we had worked on the Balesdent et al. (2018) paper published in Nature. We detail the different statistical approaches we used in the specific remarks below.

**Specific remarks:**

L76-82. Even if understandable, this paragraph is a mix of Material and Methods with objectives. I suggest authors to re-write it focusing on clear objectives. Research hypotheses are also much appreciated.

=>We agree with the reviewer and will reword the paragraph as follows:

*"This study was therefore carried out to highlight the impact of a forest to vineyard conversion on the C dynamics, while focusing on the establishment and management of a vineyard on an arenosol under a Mediterranean climate. We hypothesized that the combination of arenosol, vineyard, and conventional practices has, overall, a major impact on C stocks and remaining C dynamics in both topsoil and subsoil. To quantify our hypothesis, we chose to work on paired soils, measuring carbon contents and stocks, vertical and intra-horizon heterogeneity of carbon as measured by $^{14}C$, and correlating the C:N ratio and radiocarbon ($F^{14}C$). These parameters enabled us to: (1) determine how vine cultivation and deep ploughing impact carbon stocks and dynamics in a Mediterranean arenosol, at soil layer and entire soil profile scales and (2) use this case study to estimate, according to different calculation hypotheses, the time required for the vineyard soil to recover a C stock equivalent to that prevailing pre-cultivation."*

L81-82. It is not clear why authors applied a rate of carbon incorporation in their cultivated arenosol according to the proportions and rate put forward in the remediation study of Kazlauskaite-Jadzevice et al. (2019).
=>We acknowledge that, when presented in this way at the end of the introduction, our approach was confusing. So we will just mention in the introduction that we tested different computational assumptions, without citing, at this step, the work of Kazlauskaite-Jadzevice et al. (2019) upon which we relied. We explain the different assumptions and detail them in the last section of the discussion.

Köppen-Geigerclassification can be interesting to be used. Particularly because authors refer to it several times through the manuscript.
=>This classification was indeed used in selecting the papers underpinning the discussion: only the papers listed under "Mediterranean climate" (BSk, BWh, Cfa, Csa, Csb and Csc, see Appendix) were retained. We will add this information in the Materials and Methods section.

It should be explained in sampling whether rocks were eliminated (as well from calculations?). What happened with vegetation fragments (from roots to branches)? This should be clearly explained particularly in SOC stock studies. Is this related to the presence of less solid fragments (rocks, vegetation, etc.)?
=>Coarse material (rocks and organic matter > 2 mm) was removed with a 2 mm sieve. The remaining root tips were removed by hand. SOC stocks were calculated on the fine soil stock (STF), i.e. by removing the coarse elements from the bulk density:

$$STF = \frac{M_{samp} - (M_{samp} * EG)}{V_{samp}}$$

With STF in $g.cm^{-3}$, $M_{samp}$ in g, EG in Mass % and $V_{samp}$ in $cm^3$.

$$SOC\ stock = STF * TOC * e/10$$

With SOC stock in $t.ha^{-1}$, TOC in $g.kg^{-1}$ and e in cm.
This is now added in the "material and methods" part.

Are these results?
=>The amount of soil to be analysed with respect to 14C was defined according to the carbon content. The target was 1,000 µg of carbon for the solid source and 100 µg for the gas source, with the limitation of cumulating a maximum of 2 capsules for the solid source and 1 capsule for the gas source. One capsule can contain a maximum of 40 mg of soil. Unfortunately it was not possible to reach the 100 µg target for the deepest samples. The carbon masses used are now in the results section.

Refer to "Total Organic Content (TOC)".
=> We disagree with Reviewer #2, the carbon concentration is clearly expressed in total organic carbon: TOC. This confusion may come from line 119 where we were talking about carbon content. We will change this to total organic carbon, here at line 119 and all over the article.

Is this 0 or 5-6 to 60?
=>This was a mistake. 5 was missing. The correct depth is 50-60 cm, not 0-60 cm. This is now corrected.

L141-151. Authors should present similar depths in both treatments in order to compare them. And use p values to make sound conclusions.
=>We used a Student's t test to compare, depth by depth, the TOC between vineyard and forest soils. This test is applicable if the variances are in the same order of magnitude. We therefore performed the test on Log(TOC) to have similar orders of magnitude of the variances between vineyard and forest soils. The p-value results are:

| Depth [cm] | t-test p-value |
|------------|----------------|
| 0-5        | 0.00059        |
| 5-10       | 0.00015        |
| 10-15      | 0.00024        |
| 15-20      | 0.00028        |
| 20-30      | 0.00104        |
| 30-40      | 0.00118        |
| 40-50      | 0.00928        |
| 50-60      | 0.00100        |
| 60-70      | 0.07454        |

The p values showed a significant difference (<0.05) of TOC between forest and vineyard soils to 60 cm depth. This will be added in the article (methodology and results).

It might be good to explain why authors chose to use composite sample at these two depths (5-10 and 40-50 cm) and not others.
=> In order to minimize the 14C analysis cost (€300/sample), we opted to use composite samples for all depths: we thus obtained a mean 14C value (mean of profils A, B and C). However, the composite samples did not enable us to determine the variability in 14C at the scale of the same layer. We estimated this variability by testing it on 3 layers: a C-rich topsoil layer (5-10 cm), a C-poor subsoil layer from the vineyard ploughing horizon (40-50 cm), and a layer below the ploughing horizon for which only the soil in the vineyard was measured (50-60 cm) (in view of the 5-10 and 40-50 cm results in the forest, we did not expect that there would be any variability in the forest 50-60 cm 14C).

Section 4.3. Please include statistical analyses results that help to explain this variability.
=>Given the limited number of data, we applied a permutation test on the ratio $\frac{RMS_{forest}}{RMS_{vineyard}}$ (the residual sum of mean squares), calculated on F14C data. The RMS ratio allowed us to compare the degrees of variance between forest and vines. The permutation test allowed us to test whether the ratio result was significant or not (Manly, 2006).

At 5-10 cm depth, the observed ratio was 1.46 (≠1). We repeated 1,000 times a permutation test of the RMS ratios between forest and vines (simulation), which we then compared to the observed ratio value (Figure 1). The observed value was outside the simulated critical values with a p-value = 0 (<<0.05). This showed that the variance under vines was significantly different from the variance under forest.

[Figure]

*Figure 1 : RMS's simulation ratios in relation to the observed ratio (red) at 5-10 cm depth*

At 40-50 cm depth, the observed ratio is 0.98 (close to 1). Similarly, we repeated a permutation test 1,000 times. The observed value was within the simulated critical values (Figure 2), with a p value = 0.67 (>> 0.05). This showed that the variances under vines and under forest was not significantly different.

[Figure]

*Figure 2 : RMS's simulation ratios in relation to the observed ratio (red) at 40-50 cm depth*

In Fig.4. Why don't present both soils in one depth? Legend can be moved
The reviewer is right, this way of presenting the data is better.
=>Here is the new graph (Figure 5) that is now in the article:

[Figure]

*Figure 3 : Comparison of intra-layer F14C heterogeneity at three depths (5-10, 40-50 and 50-60 cm) in forest and vineyard soils. F14C data were obtained for profiles A (star), B (diamond), C (square), composites A+B+C (triangle) and the average of these data (round), in forest (green) and vineyard (orange) soils. Error bars represent the analytical error for profiles A, B and C and the standard deviation of the mean.*

5. Very interesting comparison.
=>Thank you.

Section 6 should be probably renamed as "Possible origin of OM". In this section there is a comparison of C:N ratios that is related to a probable origin of the OM. Authors based their discussion in Cotrufo et al., 2019. According to theses researchers, OM of plant origin shows C:N = 9.8 -17.8 and the OM of microbial origin associated with minerals C:N= 7.9-17.3. There are not great differences in these thresholds particularly when one compares results of this research with soil under forest (13 < C:N <16) and under vines (7<C:N<12). It could be in any of the two origins, don't you think?
=>The reviewer is right our approach was a bit speculative. However, we also applied a statistical approach (Student's t test) to compare the C:N between vine and forest soils. Up to 50 cm depth, the p-values were under 0.05 except for the 15-20 cm and 30-40 cm horizons, where they were less than 0.1. This result shows that there was a significant difference in C:N, with lower values in the vineyard than in the forest soils. This result tended to confirm that, at equivalent depth, the C pool remaining in the vineyard had a more marked microbial signature than the C pool in the forest soil. We will rewrite this section by changing the title as proposed by Referee #2, using the statistical results explained above, and by qualifying our statement.

| Depth [cm] | t-test p-value |
| --- | --- |
| 0-5 | 0.0255 |
| 5-10 | 0.0143 |
| 10-15 | 0.0122 |
| 15-20 | 0.0990 |
| 20-30 | 0.0098 |
| 30-40 | 0.0778 |
| 40-50 | 0.0310 |
| 50-60 | 0.4627 |
| 60-70 | 0.7696 |

6. Nothing is mentioned about Normality of data. Are these correlations made by Pearson or Spearman?
=>In our initial manuscript, we applied a simple linear regression ($R^2$=0.79). There was no normality of data (p-value=0.002), which is why the Spearman test should be preferred to the Pearson test. The Spearman correlation coefficient was r=0.78, showing that $F^{14}C$ and C:N were strongly linked by a linear relationship, which supported the regression results.

In Fig. 6. Authors should change symbols to see composite vs single samples as well. Are there any differences? I'm not sure about the independency of these data to perform correlations?
=>For a given depth, the composite samples had their own F14C measurement, so they were independent of the single samples (with regard to the F14C) and were subjected to the same errors due to the analysis. With regard to the C:N, the composite samples had a single measurement per F14C (C:N mean of the A, B and C sides), making their independence questionable. However, the results showed that the composite samples were included in the scatter plots without any preferential areas (Figure 4). So they were included in the regression. Hereafter is the new graph showing the sample types (single or composite) (Figure 4):

[Figure]

*Figure 4 : Correlation between the F14C and C:N ratio. The correlation was calculated on composite samples (F14C) and the average for the 3 profiles A, B and C (C:N), as well as on simple samples of profiles A, B and C, from the forest (green) and vineyard (orange) soils*

In addition, there was no difference in the behavior of the samples according to the depth at which they were sampled (Figure 5).

[Figure]

*Figure 5 : F14C as a function of C:N. The letters correspond to the faces of the pit, their colour to their depth (the lighter the colour, the deeper the sample).*

Therefore, we used the whole dataset to apply a linear regression and a Spearman correlation.

Are the experiments economically viable? Is the owner of vineyard willing to make this change?
=>According to a study by Pellerin et al. (2019), at the scale of France, very few of the stocking practices generate income for farmers (only 2 of the 9 studied). The economic viability of these practices will therefore depend on those chosen by the farmer, as well as on potential state aid.

Payen et al. (2022) showed that the decision to adopt stocking practices by farmers was dependent on many socioeconomic and behavioral factors (farm size, number of hired workers, attitude towards stocking practices), and on specific wine production aspects (e.g. being an independent winegrower).

We continue to work on these plots and are in contact with the wine growers of the agricultural cooperative. We hope that our work will help boost their awareness of the importance of changing agricultural practices to preserve the soil.

These issues will be addressed following the discussion on "4.7 Are arénosol a good target for the 4:1000 Initiative?".

Include depths (0-5 cm) to make it more accurate
=> OK.

L311-313. Even if the assumption of relating older age, i.e. F14C (old and stabilized carbon), to decreased C:N ratio is true, it is based on a "discussion" not completely clear (Section 6). I invite authors to re-think this part and present sound conclusions.
=>The Referee is right and, as we mentioned above, our approach was a bit speculative. However, the new statistical approaches (Spearman correlation, r = 0.78 confirmed a strong linear relationship between F14C and C:N, thereby confirming the hypothesis that an advanced age of C is related to a decrease in C:N. We will rewrite this section.

Is equation A.3. correct?
And the statistical analysis to confirm this?
=>The correct equation is:

$$As = \frac{^{14}C \; sample \; atoms}{^{12}C \; sample \; atoms}$$

In Table C1 caption, refer to Total Organic Carbon (TOC).
=>We will change it to total organic carbon.

In Table C2, C values are significantly different between A, B and C?
=>As mentioned above, amount of soil to be 14C analysed was defined according to their carbon content. The target was 1000µg of carbon for the solid source and 100µg of carbon for the gas source with the limitation of cumulating a maximum of 2 capsules for the solid source and the limitation of 1 capsule for the gas source. One capsule can contain a maximum of 40mg of soil. Unfortunately it has not yet been possible to reach the 100µg target for the deepest samples.

**References:**
Manly, B. F. J.: Randomization, Bootstrap and Monte Carlo Methods in Biology, Third Edition, CRC Press, 488 pp., 2006.

Payen, F. T., Moran, D., Cahurel, J.-Y., Aitkenhead, M., Alexander, P., and MacLeod, M.: Factors influencing winegrowers' adoption of soil organic carbon sequestration practices in France, Environmental Science & Policy, 128, 45–55, https://doi.org/10.1016/j.envsci.2021.11.011, 2022.

Sylvain Pellerin et Laure Bamière (pilotes scientifiques), Camille Launay, Raphaël Martin, Michele Schiavo, Denis Angers, Laurent Augusto, Jérôme Balesdent, Isabelle Basile-Doelsch, Valentin Bellassen, Rémi Cardinael, Lauric Cécillon, Eric Ceschia, Claire Chenu, Julie Constantin, Joël Darroussin, Philippe Delacote, Nathalie Delame, François Gastal, Daniel Gilbert, Anne-Isabelle Graux, Bertrand Guenet, Sabine Houot, Katja Klumpp, Elodie Letort, Isabelle Litrico, Manuel Martin, Safya Menasseri, Delphine Mézière, Thierry Morvan, Claire Mosnier, Jean Roger-Estrade, Laurent Saint-André, Jorge Sierra, Olivier Thérond, Valérie Viaud, Régis Grateau, Sophie Le Perchec, Isabelle Savini, Olivier Réchauchère, 2019. Stocker du carbone dans les sols français, Quel potentiel au regard de l'objectif 4 pour 1000 et à quel coût ? Synthèse du rapport d'étude, INRA (France), 114 p.

---

## Author Comment (AC4)

**Comment on soil-2021-115**

**The responses are in blue and the track-changes in green.**

**Anonymous Referee #2**
**Referee comment on "Dynamics of carbon loss from an arenosol by a forest/vineyard land use change on a centennial scale" by Solène Quéro et al., SOIL Discuss., https://doi.org/10.5194/soil-2021-115-RC2, 2022**

The manuscript "Dynamics of carbon loss from an arenosol by a forest/vineyard land use change on a centennial scale" presents, as tittle says, the results of a research about long term variations in soil organic carbon (SOC) stocks and their dynamics in a 80 cm deep Mediterranean Arenosol that had undergone a land use change from forest to vineyard over more than 100 years. According to their results a stock of 50 GtC ha-1 in the 0-30 cm forest soil horizon was reduced to 3 GtC ha-1 after long-term grape cultivation. Analyses of 14C showed that deep ploughing (50 cm) in vineyard plot redistributed the remaining carbon both vertically and horizontally. Authors concluded that this soil would have a high carbon storage potential if agricultural practices, such as grassing or organic amendment applications, were to be implemented within the framework of the 4 per 1000 Initiative.

The text denotes a considerable amount of field and laboratory work. In general, manuscript is well written (English grammar and spelling are correct). It is a very interesting research dealing with SOC stocks in soil profiles under different land uses. The natural radiocarbon (14C) abundance analyses present a significant contribution to the discussion. References are updated and they support properly introduction and discussion sections. Tables and Figures are of good quality and all necessary. However, I consider manuscript needs a MODERATE revision before being accepted for publication. It needs to consider the following remarks.

=>We thank Referee #2 for this positive evaluation and for all of the suggestions proposed to improve the paper.

**General comments:**
There are not statistical analyses supporting the data discussion. Authors are comparing values and treatments and this should be done by means of statistics.

=>We have taken this important remark of Referee#2 into account and have called upon the expertise of a statistician (Joel Chadoeuf) with whom we had worked on the Balesdent et al. (2018) paper published in Nature. We detail the different statistical approaches we used in the specific remarks below.

**Specific remarks:**

L76-82. Even if understandable, this paragraph is a mix of Material and Methods with objectives. I suggest authors to re-write it focusing on clear objectives. Research hypotheses are also much appreciated.

=>We agree with the reviewer and reworded the paragraph as follows:

"The present study was therefore carried out to highlight the impact of the long-term conversion (>100 yr) of a forest to a vineyard  on the C dynamics at the profile scale, while focusing  on an arenosol under a Mediterranean climate. We hypothesized that the combination of arenosol, vineyard and conventional practices would, overall, have a major impact on C stocks and the dynamics of C remaining  in the topsoil and subsoil. To test our hypothesis, we worked on paired soils, measuring carbon contents and stocks, vertical and intra-horizon heterogeneity of carbon, as measured by $^{14}C$, and correlating the C:N ratio and

*radiocarbon (F$^{14}$C). These parameters enabled us to: (1) determine how vineyard cultivation and deep ploughing impact carbon stocks and dynamics in a Mediterranean arenosol, at soil layer and entire soil profile scales, and (2) use this case study to estimate, according to different calculation hypotheses, the time required for the vineyard soil to recover a C stock equivalent to that prevailing pre-cultivation."*

L81-82. It is not clear why authors applied a rate of carbon incorporation in their cultivated arenosol according to the proportions and rate put forward in the remediation study of Kazlauskaite-Jadzevice et al. (2019).
=>We acknowledge that, when presented in this way at the end of the introduction, our approach was confusing. In the revised version, we explain the different assumptions and detail them in the last section of the discussion.

Köppen-Geigerclassification can be interesting to be used. Particularly because authors refer to it several times through the manuscript.
=>This classification was indeed used in selecting the papers underpinning the discussion: only the papers listed under "Mediterranean climate" (BSk, BWh, Cfa, Csa, Csb and Csc, see Appendix) were retained. We will add this information in the Materials and Methods section and in the SI.

It should be explained in sampling whether rocks were eliminated (as well from calculations?). What happened with vegetation fragments (from roots to branches)? This should be clearly explained particularly in SOC stock studies. Is this related to the presence of less solid fragments (rocks, vegetation, etc.)?
=>Coarse material (rocks and organic matter > 2 mm) was removed with a 2 mm sieve. The remaining root tips were removed by hand. SOC stocks were calculated on the fine soil stock (STF), i.e. by removing the coarse elements from the bulk density:

$$STF = \frac{M_{samp} - (M_{samp} * EG)}{V_{samp}}$$

With STF in g.cm$^{-3}$, M$_{samp}$ in g, EG in Mass % and V$_{samp}$ in cm$^3$.

$$SOC\ stock = STF * TOC * e / 10$$

With SOC stock in t.ha$^{-1}$, TOC in g.kg$^{-1}$ and e in cm.
This is now added in the "material and methods" part.

Are these results?
=>The amount of soil to be analysed with respect to $^{14}$C was defined according to the carbon content. The target was 1,000 µg of carbon for the solid source and 100 µg for the gas source, with the limitation of cumulating a maximum of 2 capsules for the solid source and 1 capsule for the gas source. One capsule can contain a maximum of 40 mg of soil. Unfortunately it was not possible to reach the 100 µg target for the deepest samples. The carbon masses used are now detailed (M&M and data table in SI).

Refer to "Total Organic Content (TOC)".
=> We disagree with Reviewer #2, the carbon concentration is clearly expressed in total organic carbon: TOC. This confusion may come from line 119 where we were talking about carbon content. We changed this to total organic carbon, here at line 119 and all over the article.

Is this 0 or 5-6 to 60?
=>This was a mistake. 5 was missing. The correct depth is 50-60 cm, not 0-60 cm. This is now corrected.

L141-151. Authors should present similar depths in both treatments in order to compare them. And use p values to make sound conclusions.

=>We used a Student's t-test to compare, depth by depth, the TOC between vineyard and forest soils. This test is applicable if the variances are in the same order of magnitude. We therefore performed the test on log$_{10}$(TOC) to have similar orders of magnitude of the variances between vineyard and forest soils. The p-value results are:

| Depth [cm] | t-test p-value |
|------------|----------------|
| 0-5        | 0.00059        |
| 5-10       | 0.00015        |
| 10-15      | 0.00024        |
| 15-20      | 0.00028        |
| 20-30      | 0.00104        |
| 30-40      | 0.00118        |
| 40-50      | 0.00928        |
| 50-60      | 0.00100        |
| 60-70      | 0.07454        |

The p-values showed a significant difference (<0.05) of TOC between forest and vineyard soils to 60 cm depth. This is added in the revised version (methodology and results).

It might be good to explain why authors chose to use composite sample at these two depths (5-10 and 40-50 cm) and not others.

=> In order to minimize the 14C analysis cost (€300/sample), we opted to use composite samples for all depths: we thus obtained a mean $^{14}C$ value (mean of profiles A, B and C). However, the composite samples did not enable us to determine the variability in $^{14}C$ at the scale of the same layer. We estimated this variability by testing it on 3 layers: a C-rich topsoil layer (5-10 cm), a C-poor subsoil layer from the vineyard ploughing horizon (40-50 cm), and a layer below the ploughing horizon for which only the soil in the vineyard was measured (50-60 cm) (in view of the 5-10 and 40-50 cm results in the forest, we did not expect that there would be any variability in the forest 50-60 cm 14C).

Section 4.3. Please include statistical analyses results that help to explain this variability.

=>Given the limited number of data, we applied a permutation test on the ratio $\frac{RMS_{forest}}{RMS_{vineyard}}$ $\frac{RMS_{vineyard}}{RMS_{forest}}$ (the residual mean squares), calculated on F$^{14}C$ data. The RMS ratio allowed us to compare the variance between forest and vines. The permutation test allowed us to test whether the ratio result was significant or not (Manly, 2006).

At 5-10 cm depth, the observed ratio was 9.16 (≠1). We repeated 1,000 times a permutation test of the RMS ratios between forest and vines (simulation), which we then compared to the observed ratio value (Figure 1). The observed value was outside the simulated critical values with a p=0 .02 (<0.05). This showed that the variance under vines was significantly different from the variance under forest.

[Figure]

[Figure]

*Figure 1 ÷: RMS's simulation ratios in relation to the observed ratio (red) at 5-10 cm depth*

At 40-50 cm depth, the observed ratio is  27.53 (≠1). Similarly, we repeated a permutation test 1,000 times. The observed value was within the simulated critical values (Figure 2), with a p  0.01 (<< 0.05). This showed that the variance under vines  was  significantly different from the variance under forest

[Figure]

[Figure]

*Figure 2: RMS's simulation ratios in relation to the observed ratio (red) at 40-50 cm depth*

In Fig.4. Why don't present both soils in one depth? Legend can be moved
The reviewer is right, this way of presenting the data is better.
=>Here is the new graph Figure 3 that is now in the article:

[Figure]

*Figure 3 : Comparison of intra-layer F¹⁴C heterogeneity at three depths (5-10, 40-50 and 50-60 cm) in forest and vineyard soils. F¹⁴C data were obtained for profiles A (star), B (diamond), C (square), composites A+B+C (triangle) and the average of these data (round), in forest (green) and vineyard (orange) soils. Error bars represent the analytical error for the profiles A, B and C and the standard deviation of for the mean.*

5. Very interesting comparison.
=>Thank you.

Section 6 should be probably renamed as "Possible origin of OM". In this section there is a comparison of C:N ratios that is related to a probable origin of the OM. Authors based their discussion in Cotrufo et al., 2019. According to theses researchers, OM of plant origin shows C:N = 9.8 -17.8 and the OM of microbial origin associated with minerals C:N= 7.9-17.3. There are not great differences in these thresholds particularly when one compares results of this research with soil under forest (13 < C:N <16) and under vines (7<C:N<12). It could be in any of the two origins, don't you think?

=>The reviewer is right our approach was a bit speculative. However, we also applied a statistical approach (Student's t-test) to compare the C:N ratios between vine and forest soils. Up to 50 cm depth, the p-values were under 0.05 except for the 15-20 cm and 30-40 cm horizons, where they were less than 0.1. This result showed that there was a significant difference in C:N, with lower values in the vineyard than in the forest soils. This result tended to confirm that, at equivalent depth, the C pool remaining in the vineyard had a more marked microbial signature than the C pool in the forest soil. We rewrote this section by changing the title as proposed by Referee #2, using the statistical results explained above, and by qualifying our statement.

| Depth [cm] | t-test p-value |
|---|---|
| 0-5 | 0.0255 |
| 5-10 | 0.0143 |
| 10-15 | 0.0122 |
| 15-20 | 0.0990 |
| 20-30 | 0.0098 |
| 30-40 | 0.0778 |
| 40-50 | 0.0310 |
| 50-60 | 0.4627 |
| 60-70 | 0.7696 |

6. Nothing is mentioned about Normality of data. Are these correlations made by Pearson or Spearman?

=>In our initial manuscript, we applied a simple linear regression ($R^2$=0.79). There was no normality of data (p=0.002), which is why the Spearman's test should be preferred to the Pearson's test. The Spearman correlation coefficient was r=0.78, showing that $F^{14}C$ and C:N were strongly linked by a linear relationship, which supported the regression results.

In Fig. 6. Authors should change symbols to see composite vs single samples as well. Are there any differences? I'm not sure about the independency of these data to perform correlations?

=>For a given depth, the composite samples had their own $F^{14}C$ measurement, so they were independent of the single samples (with regard to the $F^{14}C$) and were subjected to the same errors due to the analysis. With regard to the C:N, the composite samples had a single measurement per $F^{14}C$ (C:N mean of the A, B and C sides), making their independence questionable. However, the results showed that the composite samples were spread in the point cloud without showing any aberrant behaviour (Figure 4). So they were included in the regression. Hereafter is the new graph showing the sample types (single or composite) (Figure 4):

[Figure]

*Figure 4 : Correlation between the F14C and C:N ratio. The correlation was calculated on composite samples (F14C) and the average for the 3 profiles A, B and C (C:N), as well as on simple samples of profiles A, B and C, from the forest (green) and vineyard (orange) soils*

In addition, there was no difference in the behavior of the samples according to the depth at which they were sampled (Figure 5).

[Figure]

*Figure 5 ~: F14C as a function of C:N. The letters correspond to the faces of the pit, their colour to their depth (the lighter the colour, the deeper the sample).*

Therefore, we used the whole dataset to apply a linear regression and a Spearman correlation.

Are the experiments economically viable? Is the owner of vineyard willing to make this change?
=>According to a study by Pellerin et al. (2019), at the scale of France, very few of the stocking practices generate income for farmers (only 2 of the 9 studied). The economic viability of these practices will therefore depend on those chosen by the farmer, as well as on potential state aid.

Payen et al. (2022) showed that the decision to adopt stocking practices by farmers was dependent on many socioeconomic and behavioral factors (farm size, number of hired workers, attitude towards stocking practices), and on specific wine production aspects (e.g. being an independent winegrower).

We continue to work on these plots and are in contact with the wine growers of the agricultural cooperative. We hope that our work will help boost their awareness of the importance of changing agricultural practices to preserve the soil.

These issues is addressed at the end of the conclusion.

Include depths (0-5 cm) to make it more accurate
=> OK.

L311-313. Even if the assumption of relating older age, i.e. F14C (old and stabilized carbon), to decreased C:N ratio is true, it is based on a "discussion" not completely clear (Section 6). I invite authors to re-think this part and present sound conclusions.
=>The Referee is right and, as we mentioned above, our approach was a bit speculative. However, the new statistical approaches (Spearman correlation, r = 0.78) confirmed a strong linear relationship between $F^{14}C$ and C:N, thereby confirming the hypothesis that an advanced age of C is related to a decrease in C:N. We rewrote this section.

Is equation A.3. correct?
And the statistical analysis to confirm this?
=>The correct equation is:

$$As = \frac{^{14}C \; sample \; atoms}{^{12}C \; sample \; atoms}$$

In Table C1 caption, refer to Total Organic Carbon (TOC).
=>We changed it to total organic carbon.

In Table C2, C values are significantly different between A, B and C?
=>As mentioned above, amount of soil to be 14C analysed was defined according to their carbon content. The target was 1000 µg of carbon for the solid source and 100 µg of carbon for the gas source with the limitation of cumulating a maximum of 2 capsules for the solid source and the limitation of 1 capsule for the gas source. One capsule can contain a maximum of 40 mg of soil. Unfortunately it has not yet been possible to reach the 100 µg target for the deepest samples.

**References:**

Manly, B. F. J.: Randomization, Bootstrap and Monte Carlo Methods in Biology, Third Edition, CRC Press, 488 pp., 2006.

Payen, F. T., Moran, D., Cahurel, J.-Y., Aitkenhead, M., Alexander, P., and MacLeod, M.: Factors influencing winegrowers' adoption of soil organic carbon sequestration practices in France, Environmental Science & Policy, 128, 45–55, https://doi.org/10.1016/j.envsci.2021.11.011, 2022.

Sylvain Pellerin et Laure Bamière (pilotes scientifiques), Camille Launay, Raphaël Martin, Michele Schiavo, Denis Angers, Laurent Augusto, Jérôme Balesdent, Isabelle Basile-Doelsch, Valentin Bellassen, Rémi Cardinael, Lauric Cécillon, Eric Ceschia, Claire Chenu, Julie Constantin, Joël Darroussin, Philippe Delacote, Nathalie Delame, François Gastal, Daniel Gilbert, Anne-Isabelle Graux, Bertrand Guenet, Sabine Houot, Katja Klumpp, Elodie Letort, Isabelle Litrico, Manuel Martin, Safya Menasseri, Delphine Mézière, Thierry Morvan, Claire Mosnier, Jean Roger-Estrade, Laurent Saint-André, Jorge Sierra, Olivier Thérond, Valérie Viaud, Régis Grateau, Sophie Le Perchec, Isabelle Savini, Olivier Réchauchère, 2019. Stocker du carbone dans les sols français, Quel potentiel au regard de l'objectif 4 pour 1000 et à quel coût ? Synthèse du rapport d'étude, INRA (France), 114 p.